# VideoSAVi: Self-Aligned Video Language Models without Human Supervision

**Yogesh Kulkarni     Pooyan Fazli**
Arizona State University
{ykulka10, pooyan}@asu.edu
https://people-robots.github.io/VideoSAVi/

## Abstract

Recent advances in video-large language models (Video-LLMs) have led to significant progress in video understanding. Current preference optimization methods often rely on proprietary APIs or human-annotated captions to generate preference data (i.e., pairs of model outputs ranked by quality or alignment with human judgment), which is then used to train models for video-language alignment. This approach is both costly and labor-intensive. To address this limitation, we introduce **VideoSAVi** (**S**elf-**A**ligned **Vi**deo Language Model), a self-training pipeline that enables Video-LLMs to learn from video content without external supervision. Our approach includes a self-critiquing mechanism that identifies reasoning errors in the model's initial responses and generates improved alternatives, creating preference pairs directly from video content. VideoSAVi then applies Direct Preference Optimization (DPO) to iteratively train the model using the preference data, thus enhancing its temporal and spatial reasoning for video understanding. Experiments show that VideoSAVi delivers significant improvements across multiple benchmarks, including a +4.2 percentage point gain on MVBench, +3.9 on PerceptionTest, and +6.8 on the challenging EgoSchema dataset compared to baseline models. Our model-agnostic approach is computationally efficient, requiring only 32 frames, offering a promising direction for self-aligned video understanding without reliance on external models or annotations.

## 1 Introduction

Vision-language models (VLMs) (Liu et al., 2023; Li et al., 2023a; Radford et al., 2021; Wang et al., 2024b; Chen et al., 2024; Li et al., 2024a) have made significant strides by integrating visual perception with the reasoning capabilities of large language models (LLMs) (OpenAI, 2024; Dubey et al., 2024; Ouyang et al., 2022). These models excel in interpreting and generating contextually relevant responses through the combination of image encoders and language generation techniques. Building on this foundation, recent video-large language models (Video-LLMs) (Zhang et al., 2023; Lin et al., 2024; Li et al., 2024b; Zhang et al., 2024d; Wang et al., 2022; 2024e) incorporate temporal information by converting video frames into tokens that LLMs can process, allowing them to better analyze video content. While Video-LLMs show impressive capabilities, they generally rely on large, high-quality annotated datasets, which are resource-intensive to obtain and limit scalability.

Instruction tuning has been pivotal in advancing both VLMs and Video-LLMs (Liu et al., 2023; Brown et al., 2020; Xu et al., 2023; Wei et al., 2021; Wang et al., 2024b; Chen et al., 2024; Zhang et al., 2024c), but creating large-scale video instruction datasets is costly (Deng et al., 2024). Recent efforts to generate massive instruction datasets (up to 1.3M pairs) by distilling knowledge from proprietary models like GPT-4V (Zhang et al., 2024d) have yielded only marginal improvements. This reliance on extensive annotated data and proprietary models limits the adaptability of Video-LLMs, creating barriers to wider adoption.

Alignment-based approaches have shown promise for improving video understanding (Kulkarni & Fazli, 2025a). LLaVA-Hound (Zhang et al., 2024b) uses Direct Preference Optimization (DPO) (Rafailov et al., 2023) with text-based rankings from proprietary models, while TPO (Li et al., 2025b) focuses on temporal preference optimization by sampling negatives from video segments outside the target clip. However, both methods rely heavily on proprietary APIs or ground-truth captions, which limits their accessibility and scalability.

This raises a critical research question: *How can we train Video-LLMs to generate high-quality outputs without relying on proprietary models, ground-truth captions, or expensive human annotations while maintaining robust temporal and spatial reasoning?*

To address this challenge, we propose VideoSAVi, a novel framework requiring no external supervision beyond the model itself. Our approach focuses on post-training refinement through a four-stage process: (1) generating diverse reasoning questions and initial answers about video content, (2) self-critiquing these answers to identify factual errors and reasoning flaws, (3) revising the responses based on the self-generated feedback, and (4) leveraging these pairs for DPO to align the model towards improved reasoning capabilities.

Unlike prior approaches that depend on proprietary models or extensive annotations, VideoSAVi uniquely generates high-quality preference pairs through self-critique, identifying specific reasoning failures in both spatial relationships and temporal sequences. This self-supervision creates subtle yet unambiguous learning signals that, when optimized through DPO, enable efficient post-training refinement of capabilities already present in the model. By directly addressing the model's own weaknesses rather than conforming to external judgment, our approach delivers significant improvements across diverse benchmarks without the high computational cost of full retraining.

Our main contributions are as follows:

1. We introduce VideoSAVi, a novel self-training framework that enables Video-LLMs to reason over video content without external supervision, while maintaining computational efficiency.

2. We develop a quality-aware self-critique mechanism that generates preference pairs by focusing on both temporal and spatial reasoning, enabling comprehensive video understanding.

3. Through extensive experiments, we demonstrate that VideoSAVi delivers significant improvements across multiple benchmarks, including +4.2 percentage points on MVBench, +3.6 on NeXTQA, +3.9 on PerceptionTest, and +6.8 on EgoSchema. Moreover, VideoSAVi consistently improves various model architectures, with an average gain of +2.1 percentage points across all benchmarks.

## 2 Related Work

**Video-LLMs.** Recent Video-LLMs have made remarkable progress in multimodal understanding (Chen et al., 2024; Wang et al., 2024b; Li et al., 2024a;b; Zhang et al., 2024d). However, these models typically require massive instruction-tuning datasets and extensive computational resources for pre-training. Even with substantial training data, they often struggle with proper grounding in visual content (Fu et al., 2024; Zhou et al., 2024; Liu et al., 2024b; Kulkarni & Fazli, 2025b), particularly for videos (Wang et al., 2024a; Yao et al., 2024; Wang et al., 2024e; Song et al., 2024). While some approaches attempt to address these challenges through architecture modifications (Liu et al., 2024a; Shu et al., 2024; Li et al., 2024d; Wang et al., 2024d), specialized training objectives (Lee et al., 2024; Lan et al., 2024; Liu et al., 2024c), or inference-time adaptations (Yang et al., 2024; Hu et al., 2025; Xu et al., 2024), they typically focus on specific aspects rather than holistic alignment. In contrast, VideoSAVi introduces a post-training refinement strategy that avoids training from scratch or adding complex components. Instead of introducing entirely new capabilities, our method identifies, amplifies, and improves reasoning skills already present in the model. By leveraging self-alignment through preference optimization, VideoSAVi enables models to correct their own errors and strengthen existing temporal and spatial reasoning, without relying on external supervision or large new datasets.

**Learning from AI Feedback.** Recent work adapts preference learning techniques to video-language models to improve alignment. LLaVA-Hound-DPO (Zhang et al., 2024b) applies Direct Preference Optimization (DPO) (Rafailov et al., 2023) to video understanding, but operates mainly at the text level, without incorporating visual context. It relies on preference pairs from proprietary models like GPT-4, introducing external dependencies. Similarly, Temporal Preference Optimization (TPO) (Li et al., 2025b) focuses solely on temporal reasoning, ignores spatial relationships, and requires video captioning as an intermediate step, adding computational overhead. In contrast, VideoSAVi generates preference pairs directly from the model's own assessment of correctness, eliminating the need for external supervision or auxiliary tasks like captioning. This self-alignment approach simultaneously targets both temporal and spatial reasoning. By critiquing its own responses, the model produces training signals that directly address its specific reasoning failures rather than conforming to external judgments. This results in a more efficient and focused refinement process, targeting the model's actual weaknesses rather than pursuing general improvements.

**Self-Training.** Self-training is a powerful method for improving language model performance (Gulcehre et al., 2023; Singh et al., 2024; Huang et al., 2023; Zelikman et al., 2022; Yeo et al., 2024; Wang et al., 2024c). The core idea is for models to generate their own training data and use it to refine their performance iteratively, with notable success in enhancing reasoning and task-specific capabilities. Recent advances extend self-training to vision-language models (VLMs) (Sun et al., 2025; 2024; Deng et al., 2024) and Video-LLMs (Zohar et al., 2024). However, applying self-training to the video domain introduces unique challenges due to the complex spatiotemporal nature of videos. Existing methods often rely on expensive teacher models (Sun et al., 2024), require ground truth labels (Zohar et al., 2024), or focus on isolated aspects of video understanding. Moreover, they struggle to generate high-quality preference data that captures nuanced relationships between visual elements across time (Yin et al., 2024). Our work addresses these limitations by introducing a novel self-alignment framework tailored for video understanding. We integrate a self-critiquing mechanism with preference-based learning, enabling Video-LLMs to automatically detect and correct their own reasoning errors across spatial and temporal dimensions.

## 3 VideoSAVi

We present VideoSAVi, a novel self-aligned framework that enhances video-language models without relying on external supervision. VideoSAVi leverages existing Video-LLMs to generate challenging questions, produce and critique answers, and refine them into high-quality outputs. This self-training pipeline addresses the high costs associated with human annotations and proprietary models, while also improving the model's temporal and spatial reasoning capabilities. VideoSAVi follows a four-stage process: (1) generating reasoning questions and initial responses based on video content, (2) identifying temporal and spatial reasoning errors through self-critique, (3) revising responses based on the critique, and (4) optimizing model performance using DPO with the improved responses. Specifically, the framework prompts the baseline model, InternVL2.5 (Chen et al., 2024), along with the video input, to produce diverse questions that probe complex temporal dynamics and spatial details. The model then answers these questions, critiques its own reasoning, and generates refined responses to guide further learning. Figure 1 illustrates the overall architecture and iterative self-training pipeline of VideoSAVi.

### 3.1 Question Generation and Initial Responses

For self-alignment to be effective, generated questions must target specific reasoning skills that can be refined through preference optimization. To this end, we prompt the baseline model using an instruction template (see Appendix, Figure 8) across two core reasoning dimensions:

**Spatial Reasoning Questions.** We use prompts such as "Generate questions that require identifying spatial relationships between objects" and "Generate questions about visual details that might be easily overlooked." The resulting questions focus on spatial layouts, fine-grained visual details, and contrastive reasoning.

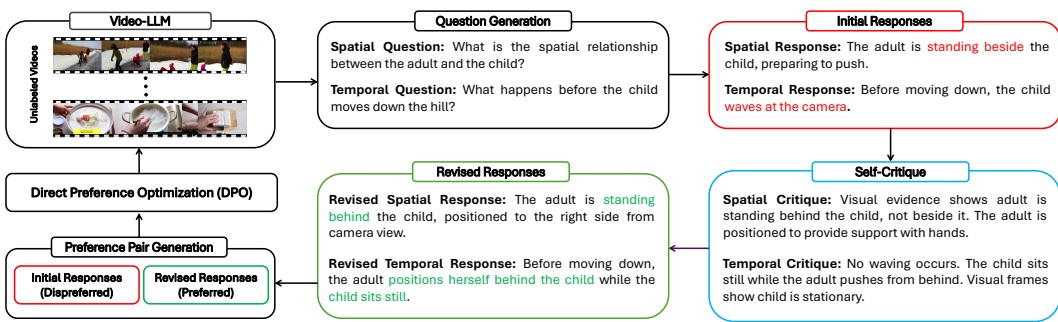

Figure 1: **Overview of VideoSAVi.** Our iterative self-training pipeline consists of **four key stages**: (1) generating diverse spatial and temporal reasoning questions and initial responses about video content, (2) applying a self-critique mechanism to identify reasoning errors and inconsistencies, (3) refining responses based on the critique feedback, and (4) training the model through DPO using paired initial (dispreferred) and revised (preferred) responses. This self-supervised cycle progressively improves VideoSAVi's reasoning capabilities without external supervision.

**Temporal Reasoning Questions.** We use prompts such as "Generate a question about the temporal sequence of events" and "Generate questions that require understanding what happened before or after specific key moments." The resulting questions challenge the model to track objects across frames and infer cause-and-effect relationships.

Despite advancements in Video-LLMs, their initial responses to such questions often reveal reasoning errors in spatial relationships, temporal sequencing, and object tracking. VideoSAVi addresses these limitations through a self-critiquing mechanism that allows the model to evaluate and improve its own reasoning without external supervision.

### 3.2 Self-Critique Mechanism

**Critique Generation.** The self-critique process begins once the model generates an initial response $a_0$ to a question-video pair $(v, q)$. It then assumes the role of its own critic by prompting itself with a specialized template (see Appendix, Figure 9) designed to elicit a critical evaluation of its reasoning:

$$c = f_{\text{critique}}(v, q, a_0; \theta). \tag{1}$$

The critique function analyzes the initial answer for inconsistencies by comparing its claims against visual evidence from the video frames. For spatial reasoning, it evaluates object positions, visibility, and attributes. For temporal reasoning, it examines event sequencing, causality, and transitions. The critique prompt encourages counterfactual reasoning by asking the model to consider alternative interpretations that may better align with the video content. This self-assessment mirrors human introspection, where evaluating one's own conclusions leads to refinement. Through this process, the model learns to balance confidence in its initial assessment with appropriate skepticism, particularly in the face of ambiguous visual input.

**Error Identification.** The critique identifies specific errors $\{e_1, e_2, ..., e_n\}$ in the initial response. Rather than being learned through a separate training process, the severity assessment function $g_\phi$ is implicitly encoded in the model's parameters via preference optimization:

$$\lambda_i = g_\phi(e_i). \tag{2}$$

The model implicitly learns to assess error severity by developing an understanding, refined through preference optimization, of how different types of mistakes affect response quality.

It categorizes errors as either critical (e.g., factual inaccuracies, logical contradictions) or minor (e.g., imprecisions, stylistic issues) based on their influence on overall correctness.

The critique specifically targets common failure modes in video understanding, including hallucinations of non-existent objects or actions, errors in temporal ordering that misrepresent event sequences (Li et al., 2025a), incorrect causal attributions, misinterpretations of spatial relationships, and failures in tracking entities across frames.

### 3.3 Response Refinement

The refinement function is implemented via a prompt template rather than a separately parameterized module. The initial response $a_0$ and the generated critique $c$ are combined into a new prompt (see Appendix, Figure 10) that instructs the model to produce an improved answer:

$$a_1 = \pi_\theta(v, q, a_0, c). \tag{3}$$

This refinement prompt includes instructions such as "Consider the following critique of your previous answer and provide an improved response that addresses these issues." The model then generates a refined answer $a_1$ that incorporates the feedback from the self-critique.

Several principles guide the refinement process: maintaining factual correctness by ensuring all claims are directly observable in the video, preserving useful information from the original response while correcting errors, avoiding the introduction of new speculation that is not supported by visual evidence, and ensuring logical consistency throughout the response.

### 3.4 Preference Pair Creation and Optimization

The initial (dispreferred) response $a_0$ and the revised (preferred) response $a_1$ form a preference pair for DPO. These pairs are then used to train the model with the following loss function:

$$\mathcal{L}_{\text{DPO}}(\theta) = -\mathbb{E}_{(v,q,a_0,a_1)\sim\mathcal{D}}\left[\log\sigma\left(\beta\log\frac{\pi_\theta(a_1|v,q)}{\pi_{\text{ref}}(a_1|v,q)} - \beta\log\frac{\pi_\theta(a_0|v,q)}{\pi_{\text{ref}}(a_0|v,q)}\right)\right], \tag{4}$$

where $\pi_\theta$ is the model policy being trained, $\pi_{\text{ref}}$ is a reference policy derived from the initial model, and $\beta$ is a scaling hyperparameter. This objective encourages the model to assign higher probability to refined, more accurate responses ($a_1$) over the initial, flawed ones ($a_0$).

Through repeated iterations of the entire pipeline (i.e., initial response generation, self-critiquing, refinement, and preference optimization), VideoSAVi progressively improves its ability to generate accurate responses and to identify and correct its own reasoning errors. This iterative process creates a self-improving cycle that enhances both spatial and temporal reasoning capabilities without requiring external supervision.

## 4 Experiments and Evaluations

For training data, we sample a diverse collection of 4,000 videos from three sources: 2,000 from Star (Wu et al., 2021), and 1,000 each from Vidal (Zhu et al., 2024) and WebVid (Bain et al., 2021). Our self-supervised pipeline yields a comprehensive dataset of 24,000 preference pairs (6 pairs per video: 3 spatial reasoning + 3 temporal reasoning × 4,000 videos). To ensure fair evaluation, we decontaminate the training data by removing any videos that overlap with the evaluation benchmarks. VideoSAVi uses state-of-the-art InternVL2.5 (Chen et al., 2024) as its backbone and is optimized using the SWIFT (Zhao et al., 2024) framework. We employ LoRA (Hu et al., 2021) with $\alpha = 8$ (and rank $r = 8$), and DPO with temperature $\beta = 0.1$. All training and evaluation are conducted on two NVIDIA L40S GPUs (48GB), with

| Model | TempCompass | PerceptionTest | NeXTQA | MVBench | EgoSchema | LongVideoBench |
|---|---|---|---|---|---|---|
| *(1) Baseline Models* | | | | | | |
| InternVL2.5 (Chen et al., 2024) | 68.3 | 62.2 | 77.0 | 69.8 | 52.0 | 57.8 |
| + SFT | 68.5 | 63.0 | 77.5 | 70.2 | 53.0 | 58.0 |
| + SFT⁺ | 68.7 | 64.5 | 78.3 | 71.6 | 54.5 | 58.1 |
| + Hound-DPO (Zhang et al., 2024b) | 66.8 | 61.0 | 74.8 | 64.2 | 48.5 | 54.3 |
| + TPO (Li et al., 2025b) | 68.2 | 62.0 | 77.2 | 68.8 | 52.8 | 58.1 |
| *(2) State-of-the-Art Models* | | | | | | |
| VideoLLaMA2⁺ (Cheng et al., 2024) | 43.4 | 51.4 | - | 54.6 | 51.7 | - |
| Kangaroo⁺ (Liu et al., 2024a) | - | - | - | 61.0 | - | 54.8 |
| LLaVA-NeXT-Video⁺ (Zhang et al., 2024c) | 53.0 | 48.8 | 53.5 | 53.1 | - | 49.1 |
| LLaVA-NeXT-Interleave (Li et al., 2024b) | 54.1 | 51.2 | 67.0 | 46.5 | 51.0 | 44.8 |
| Qwen2-VL (Wang et al., 2024b) | 68.9 | 62.3 | 75.7 | 64.9 | 59.2 | 55.6 |
| LLaVA-OneVision (Li et al., 2024a) | 64.5 | 57.1 | 79.3 | 56.7 | 64.0 | 56.3 |
| LLaVA-Video (Zhang et al., 2024d) | 66.4 | 67.9 | 74.2 | 58.6 | 57.6 | 58.2 |
| *(3) Preference-Optimized Models* | | | | | | |
| LLaVA-Hound-DPO (Zhang et al., 2024b) | 55.5 | 45.1 | 61.6 | 36.6 | 36.1 | 36.7 |
| i-SRT (Ahn et al., 2024) | 56.0 | 47.0 | 63.0 | 36.3 | 46.2 | 38.2 |
| LLaVA-Video-TPO (Li et al., 2025b) | 66.6 | 66.3 | 77.8 | 56.7 | 58.0 | 58.3 |
| **VideoSAVi** | **69.1**₊₀.₈ | 66.1₊₃.₉ | **80.6**₊₃.₆ | **74.0**₊₄.₂ | 58.8₊₆.₈ | 59.8₊₂.₀ |

Table 1: **Comprehensive evaluation of VideoSAVi against leading video understanding models**. Best scores are in **bold**, and second-best scores are underlined. For VideoSAVi, performance improvements over InternVL 2.5 are shown as subscripts (reported as absolute gains in percentage points). All results except those marked with ⁺ are reproduced using LMMs-Eval (Zhang et al., 2024a).

a maximum of 32 frames per video to avoid CUDA out-of-memory errors. The full training process consists of four iterations of self-training. Each iteration takes approximately 48 GPU hours (24 hours for generating preferences and 24 hours for DPO training), resulting in a total of ~192 GPU hours across all iterations. For evaluation, we use LMMs-Eval (Zhang et al., 2024a) to ensure consistent comparison with prior work. Additional experiments and analyses are provided in the appendix, including: iteration-wise performance (§A.1), preference data composition (§A.2), training data composition (§A.3), DPO training dynamics (§A.4), qualitative examples (§A.6), dataset samples (§A.7), critique examples (§A.8), and prompt design (§A.9).

**Benchmarks.** We evaluate VideoSAVi on a range of general video understanding benchmarks, each targeting a specific reasoning skill: MVBench for multi-task reasoning (Li et al., 2024c), PerceptionTest for visual perception (Patraucean et al., 2024), TempCompass for temporal understanding (Liu et al., 2024b), and NeXTQA for compositional reasoning (Xiao et al., 2021). To assess performance on long-form video understanding, we use EgoSchema, which features 3-minute egocentric videos (Mangalam et al., 2023), and LongVideoBench, which includes hour-long videos across diverse tasks (Wu et al., 2024).

## 4.1 Results

We compare VideoSAVi with (1) baseline models built on InternVL2.5 (Chen et al., 2024), (2) current state-of-the-art models, and (3) models enhanced through preference optimization. Table 1 presents the evaluation results.

**VideoSAVi achieves substantial gains over foundation models across all benchmarks.** Compared to its foundation model, InternVL2.5 (Chen et al., 2024), VideoSAVi shows significant improvements across all benchmarks: +0.8 percentage points on TempCompass, +3.9 on PerceptionTest, +3.6 on NeXTQA, +4.2 on MVBench, +6.8 on EgoSchema, and +2.0 on LongVideoBench. Supervised fine-tuning (SFT) on VInstruct (Maaz et al., 2023) and our aligned SFT⁺ with preferred responses show modest improvements, but neither matches our model's generalization capabilities. Notably, VideoSAVi overcomes the limitations of Hound-DPO (Zhang et al., 2024b), which demonstrates that purely text-based ranking of preferences is fundamentally inadequate for video understanding, leading to severe performance degradation on MVBench (-5.6) and EgoSchema (-3.5).

**VideoSAVi outperforms previous state-of-the-art on key benchmarks through self-alignment.** Our approach surpasses Qwen2-VL (Wang et al., 2024b) by +9.1 percentage points on MVBench achieving 74.0% accuracy, demonstrating significant improvements

| Error Type | Iter1 | Iter2 | Iter3 | Iter4 |
|---|---|---|---|---|
| Factual Inaccuracy | 287 | 198 | 127 | 73 |
| Temporal Ordering | 243 | 162 | 108 | 52 |
| Spatial Relationship | 208 | 147 | 92 | 48 |
| Object Hallucination | 169 | 118 | 71 | 33 |
| Causal Reasoning | 132 | 91 | 58 | 27 |
| Detail Omission | 118 | 87 | 44 | 16 |
| **Total** | **1157** | **803** | **500** | **249** |

| Test Condition | VideoSAVi | TPO | H-DPO |
|---|---|---|---|
| Training Dist. | 74.5 | 71.2 | 69.8 |
| Cross-Question | $70.2_{-4.3}$ | $62.5_{-8.7}$ | $56.2_{-13.6}$ |
| Cross-Video | $69.8_{-4.7}$ | $61.8_{-9.4}$ | $54.1_{-15.7}$ |
| Full OOD | $68.7_{-5.8}$ | $59.4_{-11.8}$ | $51.3_{-18.5}$ |
| Adversarial | $67.9_{-6.6}$ | $57.8_{-13.4}$ | $48.5_{-21.3}$ |
| Compositional | $68.3_{-6.2}$ | $58.6_{-12.6}$ | $49.2_{-20.6}$ |
| **Avg. Gen. Gap** | **-6.4** | **-13.5** | **-17.9** |

Table 2: **Reasoning Errors Across Iterations**. Table 3: **Generalization capability across increasingly challenging test conditions**.

over existing methods. On NeXTQA, VideoSAVi outperforms the previous best, LLaVA-OneVision (Li et al., 2024a), by +1.3 percentage points. While LLaVA-Video (Zhang et al., 2024d) maintains an edge on PerceptionTest (67.9% vs. 66.1%), and LLaVA-OneVision leads on EgoSchema (64.0% vs. 58.8%), VideoSAVi demonstrates stronger generalization across the full set of benchmarks.

**VideoSAVi redefines preference optimization for robust video understanding.** Previous preference-optimized approaches show inconsistent performance across benchmarks. LLaVA-Hound-DPO (Zhang et al., 2024b) and i-SRT (Ahn et al., 2024) suffer large performance drops, and TPO (Li et al., 2025b) fails to improve temporal understanding despite its specific focus. In contrast, VideoSAVi delivers consistent improvements, outperforming TPO by +1.5 percentage points on LongVideoBench and reaching 59.8%. These findings challenge the assumption that preference optimization necessarily compromises consistency. Our self-alignment approach demonstrates robust performance across all dimensions of video understanding, maintaining strong capabilities in both temporal reasoning and spatial comprehension, where previous methods show significant variability.

## 4.2 Ablation Studies

**Reasoning Errors Across Iterations.** To address concerns about potential error reinforcement in our self-critique approach, we conduct an error propagation analysis across four iterations of VideoSAVi (Table 2). We use 1200 video-question pairs generated by iteration 1 and have each subsequent iteration (2, 3, and 4) independently answer the same questions without access to previous responses. GPT-4o (Hurst et al., 2024) evaluates all answers using the prompt provided in Figure 12 (see Appendix). This setup tests for the risk of "self-delusion," where a self-improving method might reinforce its own incorrect beliefs instead of correcting them. The results show a consistent and substantial reduction in errors, with total mistakes decreasing by 78.4%, from 1157 in iteration 1 to 249 in iteration 4. This reduction closely aligns with our iteration-wise performance improvements (Appendix §A.1), confirming that VideoSAVi learns to correct its reasoning rather than reinforcing errors.

**Memorization vs. Generalization.** Table 3 evaluates VideoSAVi's ability to generalize beyond its training distribution. We design a comprehensive test bed (detailed methodology in Appendix §A.5) with increasingly challenging conditions: novel question formulations (Cross-Question), unseen videos (Cross-Video), new domains (Full OOD), deliberately challenging inputs (Adversarial), and cases requiring integrative reasoning (Compositional). VideoSAVi demonstrates exceptional robustness with an average generalization gap of only -6.4 percentage points. This resilience is particularly evident in the most challenging scenarios, in which VideoSAVi maintains 68.3% accuracy on compositional questions and 67.9% on adversarial examples. In contrast, TPO (Li et al., 2025b) shows a larger average gap of -13.5 percentage points, particularly struggling with adversarial reasoning (-13.4), while Hound-DPO (H-DPO) (Zhang et al., 2024b) shows the largest generalization gap (-17.9 percentage points), with performance dropping by nearly 21 percentage points on adversarial examples. The small performance variance across conditions confirms that our self-alignment approach develops genuine reasoning capabilities rather than memorizing patterns from training data.

| Critic Type | MVB | ES | LVB |
|---|---|---|---|
| InternVL2.5 (Baseline) | 69.8 | 52.0 | 57.8 |
| SFT$^{\text{Hound-DPO}}$ | 64.2 | 48.5 | 54.3 |
| SFT$^{\text{TPO}}$ | 68.8 | 52.8 | 58.1 |
| GPT-4o | 72.0 | 57.0 | 59.0 |
| **VideoSAVi** | **74.0** | **58.8** | **59.8** |

Table 4: **Effect of different critics on model performance**. MVB: MVBench, ES: EgoSchema, LVB: LongVideoBench.

| Model | MVB | ES | LVB |
|---|---|---|---|
| *Smaller Models with Preference Learning* | | | |
| InternVL (1B) | 63.5 | 39.6 | 45.4 |
| + VideoSAVi | 64.8$_{+1.3}$ | 43.8$_{+4.2}$ | **48.3**$_{+2.9}$ |
| InternVL (2B) | **65.9** | **44.7** | 48.0 |
| *Parameter Scaling Comparison* | | | |
| InternVL (2B) | 65.9 | 44.7 | 48.0 |
| + VideoSAVi | 67.5$_{+1.6}$ | 49.4$_{+4.7}$ | **52.3**$_{+4.3}$ |
| InternVL (4B) | **68.5** | **55.3** | 51.9 |

Table 5: **Preference Learning vs. Model Scaling**. MVB: MVBench, ES: EgoSchema, LVB: LongVideoBench.

**Externally Trained Critics vs. Self-Critiquing.** Table 4 compares our self-critiquing approach to external critique methods. We evaluate: (1) SFT with GPT-4o-generated critiques of preference pairs from existing methods (SFT$^{\text{Hound-DPO}}$ and SFT$^{\text{TPO}}$), (2) GPT-4o as a critique, and (3) VideoSAVi's self-critiquing framework. Despite leveraging sophisticated external critique generation (prompt in Appendix, Figure 11), both SFT variants underperform. SFT$^{\text{Hound-DPO}}$ suffers a significant drop (-5.6 percentage points on MVBench) while SFT$^{\text{TPO}}$ shows only minimal improvement. GPT-4o-based critiquing performs better (+2.2 percentage points on MVBench), yet VideoSAVi's self-critiquing consistently achieves superior

| Benchmark | GPT-4o |
|---|---|
| TempCompass | 73.8 |
| PerceptionTest | 72.1 |
| NeXTQA | 88.1 |
| MVBench | 64.6 |
| EgoSchema | 66.2 |
| LongVideoBench | 66.7 |

Table 6: GPT-4o performance on video understanding benchmarks.

results across all benchmarks (+4.2 percentage points on MVBench, +6.8 on EgoSchema). While GPT-4o achieves strong results overall (Table 6), its critiques introduce out-of-distribution, off-policy preferences that misalign with the model's training distribution, diminishing DPO's effectiveness due to KL-divergence constraints. In contrast, VideoSAVi's on-policy critiques remain within the model's natural output distribution, enabling more stable and effective preference learning. These results highlight the advantage of integrating critique generation directly into the learning loop over relying on external critics.

**Parameter Scaling vs. Preference Learning.** Table 5 illustrates the comparative efficiency of preference learning vs. parameter scaling for enhancing video understanding. The results show that VideoSAVi consistently delivers substantial improvements across all model sizes and benchmarks. While larger models generally achieve better performance, VideoSAVi offers complementary and cost-effective improvements. For instance, a 1B parameter model with VideoSAVi (48.3% on LongVideoBench) achieves comparable performance to a baseline 2B model (48.0%). This efficiency advantage becomes particularly significant when considering the computational resources required for scaling parameters vs. our approach, which requires only self-generated preference pairs.

**Human Evaluation.** To validate the quality of self-generated preference pairs, we conduct a rigorous human evaluation with 6 external evaluators across 516 videos (261 temporal and 255 spatial). Evaluators are shown videos along model-generated questions and preference pairs without revealing the reasoning category. A pair is considered correct if and only if: (1) the question is answerable from video content, (2) the preferred response is accurate, and (3) the dispreferred response contains clear errors. This strict protocol reveals strong performance for both spatial relationships (71.3%) and temporal ordering (67.1%), with an overall quality of 69.2%. These results are particularly notable given that the preference learning process is fully self-supervised, operating without human intervention or external models. This confirms that our self-critiquing mechanism identifies genuine reasoning errors rather than arbitrary preferences, demonstrating that self-alignment can generate high-quality training signals for complex video understanding tasks.

| Model | TempCompass | PerceptionTest | NeXTQA | MVBench | EgoSchema | LongVideoBench |
|---|---|---|---|---|---|---|
| VideoLLaVA (Lin et al., 2024) | 34.3 | 42.6 | 58.4 | 34.1 | 18.8 | 39.1 |
| + VideoSAVi | 36.8 +2.5 | 45.9 +3.3 | 63.2 +4.8 | 38.7 +4.6 | 23.5 +4.7 | 41.8 +2.7 |
| LLaVA-NeXT-Interleave (Li et al., 2024b) | 53.2 | 51.0 | 67.3 | 46.5 | 51.0 | 44.8 |
| + VideoSAVi | 55.6 +2.4 | 54.4 +3.4 | 71.2 +3.9 | 50.3 +3.8 | 54.6 +3.6 | 47.9 +3.1 |
| LLaVA-OneVision (Li et al., 2024a) | 64.1 | 57.5 | 79.3 | 56.7 | 64.0 | 56.3 |
| + VideoSAVi | 66.3 +2.2 | 60.8 +3.3 | 80.9 +1.6 | 59.9 +3.2 | 65.7 +1.7 | 58.7 +2.4 |
| Qwen2-VL (Wang et al., 2024b) | 68.9 | 62.1 | 75.6 | 64.9 | 59.2 | 55.6 |
| +VideoSAVi | 68.4 −0.5 | 65.4 +3.3 | 78.8 +3.2 | 68.3 +3.4 | 62.8 +3.6 | 58.1 +2.5 |
| Qwen2.5VL (Bai et al., 2025) | 72.3 | 68.6 | 75.8 | 65.2 | 60.9 | 60.7 |
| +VideoSAVi | 72.4 +0.1 | 68.9 +0.3 | 75.9 +0.1 | 65.4 +0.2 | 61.0 +0.1 | 60.7 +0.0 |
| Qwen2.5-Omni (Xu et al., 2025) | 69.6 | 66.6 | 75.2 | 63.0 | 61.8 | 56.9 |
| +VideoSAVi | 69.9 +0.3 | 66.9 +0.3 | 75.6 +0.4 | 64.3 +1.3 | 61.9 +0.1 | 57.2 +0.3 |

Table 7: **Model-agnostic improvements from VideoSAVi**.

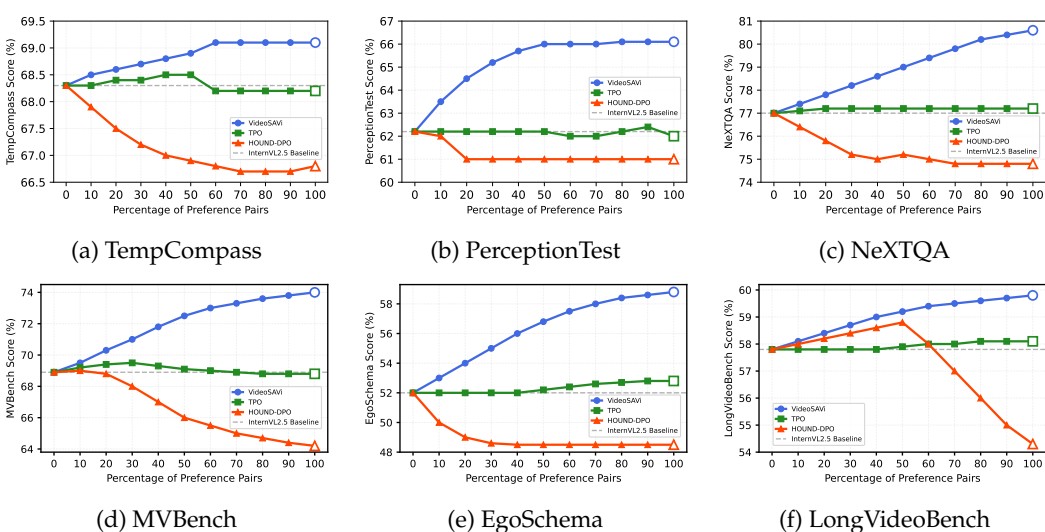

(a) TempCompass    (b) PerceptionTest    (c) NeXTQA

(d) MVBench    (e) EgoSchema    (f) LongVideoBench

Figure 2: **Impact of preference data size on model performance**.

**Model-Agnostic Improvements.** Table 7 demonstrates the broad applicability of our approach across diverse architectures. VideoSAVi consistently enhances performance, delivering substantial improvements on nearly all benchmarks, regardless of the base model. The gains are particularly notable for VideoLLaVA (Lin et al., 2024), with improvements of +4.8 percentage points on NeXTQA, +4.7 on EgoSchema, and +4.6 on MVBench, showing that models with greater headroom benefit most from our preference optimization. For mid-tier models like LLaVA-NeXT-Interleave (Li et al., 2024b), VideoSAVi provides well-balanced improvements (+3.9 on NeXTQA, +3.8 on MVBench, +3.6 on EgoSchema), strengthening both spatial and temporal reasoning capabilities. Notably, VideoSAVi also enhances top-performing models, improving LLaVA-OneVision (Li et al., 2024a) by +1.6 on NeXTQA, reaching an impressive 80.9% and boosting Qwen2-VL (Wang et al., 2024b) across five benchmarks. For state-of-the-art models like Qwen2.5VL (Bai et al., 2025) and Qwen2.5-Omni (Xu et al., 2025) that have already undergone post-training refinements using DPO, VideoSAVi still delivers consistent but more modest improvements, with notable gains on MVBench (+0.2 and +1.3 respectively), demonstrating our method's effectiveness even when applied to already-optimized models. These consistent cross-architecture improvements, averaging +2.1 percentage points across all models and benchmarks, confirm that our self-aligning preference learning framework addresses fundamental challenges in video understanding rather than exploiting model-specific characteristics.

**Impact of Preference Data Size.** Figure 2 demonstrates how VideoSAVi's self-alignment pipeline consistently improves performance across all benchmarks as the percentage of preference pairs from the final (fourth) iteration of self-training increases. In contrast, TPO (Li et al., 2025b) shows minimal improvements or plateaus (particularly on EgoSchema), and

| Ability | Benchmark | Base | +VideoSAVi |
|---|---|---|---|
| Multimodal | MMMU | 53.1 | $\mathbf{53.2}_{+0.1}$ |
| Document | DocVQA | 91.3 | $\mathbf{91.9}_{+0.6}$ |
| OCR/Text | TextVQA | 79.2 | $\mathbf{79.8}_{+0.6}$ |
| Reasoning | GSM8k | $\mathbf{77.4}$ | $76.9_{-0.5}$ |
| Video Cap. | VidChat | 2.73 | $\mathbf{3.0}_{+0.27}$ |
| Hallucination | POPE | 90.1 | $\mathbf{90.3}_{+0.2}$ |

Table 8: **VideoSAVi preserve core capabilities**.

| Model | MVB | ES | LVB |
|---|---|---|---|
| InternVL2.5 | 69.8 | 52.0 | 57.8 |
| + Noisy Pref. | $71.2_{+1.4}$ | $57.1_{+5.1}$ | $58.2_{+0.4}$ |
| + **VideoSAVi** | $\mathbf{74.0}_{+4.2}$ | $\mathbf{58.8}_{+6.8}$ | $\mathbf{59.8}_{+2.0}$ |

Table 9: **Robustness to preference data noise**. MVB: MVBench, ES: EgoSchema, LVB: LongVideoBench.

Hound-DPO (Zhang et al., 2024b) experiences notable performance drops after initial improvements. VideoSAVi's robustness and steady progress throughout training stems from its self-critiquing mechanism that targets challenging reasoning cases by effectively identifying subtle temporal inconsistencies and spatial reasoning errors. The steep improvement curve on EgoSchema (+6.8 percentage points) highlights how our approach excels at complex egocentric understanding tasks, while the consistent gains on LongVideoBench demonstrate effective temporal alignment even when competing methods struggle with longer-form content.

**Preservation of Core Capabilities.** To address concerns about potential overfitting during self-training, we evaluate VideoSAVi's performance across a diverse set of benchmarks, assessing the backbone model's pre-training performance (Chen et al., 2024). As shown in Table 8, VideoSAVi largely maintains or improves performance across all domains, preserving multimodal understanding (MMMU (Yue et al., 2024): +0.1 percentage points), document comprehension (DocVQA (Mathew et al., 2021): +0.6), and OCR reasoning (TextVQA (Mathew et al., 2021): +0.6), while enhancing video captioning (VideoChat-GPT (Maaz et al., 2024): +0.27) and reducing hallucination (POPE (Li et al., 2023b): +0.2). This preservation is achieved by freezing the vision encoder and projector while applying low-rank LoRA adapters (rank 8) to update only 0.1% of language model parameters, retaining original visual feature extraction while enhancing language model reasoning over visual features.

**Robustness to Noisy Preferences.** To test VideoSAVi's robustness to noisy preference data, we deliberately flip the labels of 30% of our 24,000 preference pairs, introducing substantial incorrect training signals. Table 9 shows that despite this significant noise, VideoSAVi with noisy preferences still consistently outperforms the baseline (+1.4 percentage points on MVBench, +5.1 on EgoSchema, +0.4 on LongVideoBench), while the full VideoSAVi pipeline maintains superior performance. This robustness stems from DPO's inherent regularization through KL-divergence constraints and the on-policy nature of our self-generated preference pairs, which stay within the model's natural output distribution and avoid instability associated with external preference data.

# 5 Conclusion

VideoSAVi is a novel self-aligned framework that employs a self-critiquing mechanism to detect and correct spatial and temporal reasoning failures in video-language models. It generates high-quality preference pairs directly from video content without requiring external supervision. Extensive experiments show substantial performance improvements over baseline models on a comprehensive set of benchmarks, demonstrating robust generalization across diverse test conditions. The method is computationally efficient and model-agnostic, enabling smaller models to outperform larger baselines through remarkable parameter efficiency. Future work will explore self-rewarding mechanisms and automatic hallucination detection frameworks, further enhancing video understanding reliability without reliance on external annotations or proprietary models.

## Reproducibility Statement

To ensure the reproducibility of our research, we rely on publicly available datasets and frameworks. Our training videos are sourced from open datasets, including Star (Wu et al., 2021), Vidal (Zhu et al., 2024), and WebVid (Bain et al., 2021). We implement our approach using the SWIFT training framework (Zhao et al., 2024) and evaluate it using the LMMs-Eval (Zhang et al., 2024a), both of which are open source. Further, we will release our final trained model checkpoints and the complete set of model-generated preference pairs. A list of all prompts used in our implementation can be found in Appendix §A.9.

## Acknowledgments

This research was supported by the National Eye Institute (NEI) of the National Institutes of Health (NIH) under award number R01EY034562. The content is solely the responsibility of the authors and does not necessarily represent the official views of the NIH.

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

# A Appendix

## A.1 Iteration-wise Analysis

Figure 3 demonstrates the iteration-wise performance improvements of VideoSAVi across six video understanding benchmarks. Each iteration represents a complete cycle of our self-alignment pipeline, where the model (1) generates questions targeting specific reasoning capabilities and produces initial responses, (2) self-critiques these responses to identify reasoning errors, (3) creates revised responses based on critique feedback, and (4) uses the revised and initial responses to form preference pairs for DPO training. The consistent upward trajectory across all benchmarks validates our hypothesis that self-critiquing enables effective self-training without external supervision. Most notably, we observe substantial gains on tasks requiring complex reasoning: NeXTQA (+3.6 percentage points), MVBench (+5.1), and EgoSchema (+6.8). These improvements are particularly significant since EgoSchema features ego-centric videos with complex temporal relationships, while MVBench demands both fine-grained spatial understanding and temporal reasoning. The performance gains exhibit different patterns across benchmarks. TempCompass shows more modest improvements (+0.8), likely because our baseline model already performs competitively on this benchmark. In contrast, PerceptionTest demonstrates steady improvement (+3.9), indicating that VideoSAVi's self-critiquing effectively addresses perceptual reasoning errors. For long-form videos (LongVideoBench), we observe a +2.0 improvement, confirming that our approach can enhance extended temporal reasoning capabilities. The iteration-wise analysis reveals that the most significant improvements occur in earlier iterations (especially iterations 1-2). This pattern suggests that VideoSAVi quickly identifies and corrects the most critical reasoning errors, followed by more minor refinements in subsequent iterations. Importantly, unlike methods that rely on external supervision or proprietary models, VideoSAVi achieves these improvements entirely through self-refinement, demonstrating the usability of self-aligned video understanding.

## A.2 Preference Composition Analysis

The decomposition of preference types in Table 10 reveals distinct task-specific effectiveness patterns. Spatial-only preferences lead to significant improvements on perception-intensive

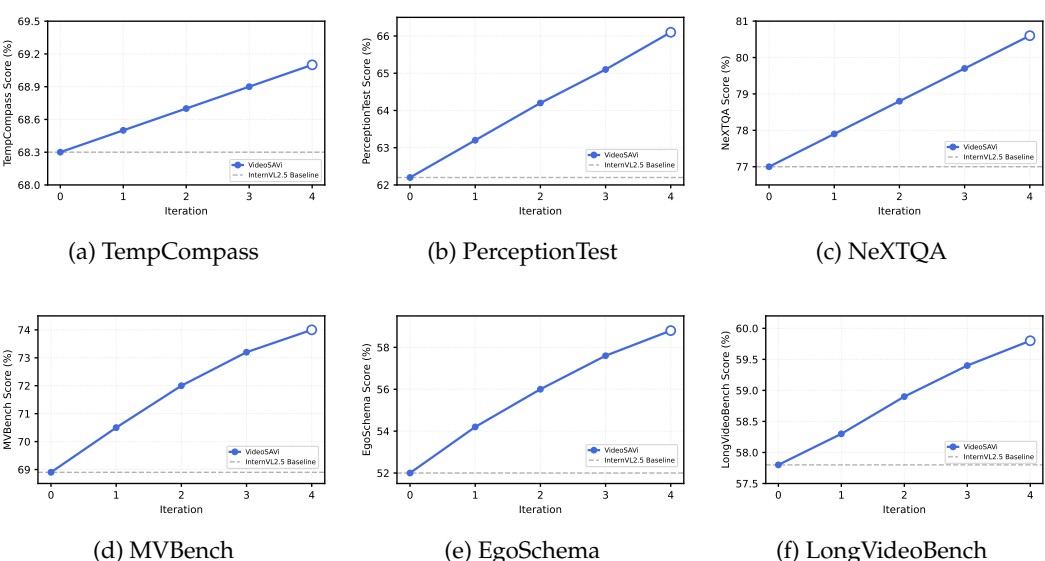

(a) TempCompass      (b) PerceptionTest      (c) NeXTQA

(d) MVBench      (e) EgoSchema      (f) LongVideoBench

Figure 3: **Iteration-wise performance improvement of VideoSAVi** across six video understanding benchmarks. Each iteration refines the model through self-critiquing and preference optimization, demonstrating consistent performance gains without external supervision.

| Preference Type | Benchmark Performance | | | | | |
|---|---|---|---|---|---|---|
| | TempCompass | PerceptionTest | NeXTQA | MVBench | EgoSchema | LongVideoBench |
| *Baseline* | | | | | | |
| InternVL2.5 | 68.3 | 62.2 | 77.0 | 68.9 | 52.0 | 57.8 |
| *Preference Composition* | | | | | | |
| Spatial Only | $68.8_{+0.5}$ | $65.5_{+3.3}$ | $78.9_{+1.9}$ | $72.2_{+3.3}$ | $55.3_{+3.3}$ | $58.4_{+0.6}$ |
| Temporal Only | $69.0_{+0.7}$ | $63.1_{+0.9}$ | $79.5_{+2.5}$ | $70.5_{+1.6}$ | $56.1_{+4.1}$ | $59.2_{+1.4}$ |
| **Full (Spatial+Temporal)** | $\mathbf{69.1}_{+0.8}$ | $\mathbf{66.1}_{+3.9}$ | $\mathbf{80.6}_{+3.6}$ | $\mathbf{74.0}_{+5.1}$ | $\mathbf{58.8}_{+6.8}$ | $\mathbf{59.8}_{+2.0}$ |

Table 10: **Preference Composition Analysis of VideoSAVi**. We evaluate the contribution of spatial and temporal preference types across six benchmarks. Improvements over the baseline are shown as subscripts. While each type contributes to performance gains, their complementary combination in the full VideoSAVi yields consistently superior results, demonstrating the importance of addressing both spatial and temporal reasoning capabilities for comprehensive video understanding. Notably, spatial preferences provide larger gains on perception-focused tasks, while temporal preferences excel on longer-form video understanding, but the integration of both creates the strongest overall results.

benchmarks (PerceptionTest: +3.3 percentage points, MVBench: +3.3), demonstrating that targeting visual reasoning errors substantially enhances scene understanding capabilities. These gains align with the inherent spatial reasoning demands of these benchmarks, which require precise object localization and attribute recognition.

Temporal-only preferences demonstrate complementary strengths, excelling on benchmarks with extended temporal dependencies (EgoSchema: +4.1, LongVideoBench: +1.4, NeXTQA: +2.5). The effectiveness on EgoSchema is particularly noteworthy, as egocentric videos contain complex action sequences requiring precise temporal ordering and causality inference. Temporal preference learning effectively addresses common temporal reasoning failures, including sequence ordering errors and causal misattributions.

The integration of both preference types yields strong results on several benchmarks. For MVBench, the full model achieves a +5.1 improvement, exceeding the sum of individual components (+3.3 spatial, +1.6 temporal). This non-linear gain suggests that joint optimization addresses compound reasoning errors that span both dimensions, particularly for tasks requiring spatiotemporal reasoning (e.g., tracking object state changes over time).

Analysis of performance differentials reveals benchmark-specific optimization patterns. On perception-focused tasks (PerceptionTest, MVBench), spatial preferences contribute disproportionately to the full model's gains, while on temporally complex benchmarks (EgoSchema, LongVideoBench), temporal preferences provide complementary strengths that enhance the combined model. This task-specific contribution pattern validates our approach of targeting both reasoning dimensions simultaneously rather than optimizing for a single aspect.

## A.3 Training Dataset Composition Analysis

To investigate the impact of dataset diversity on self-alignment performance, we conducted an ablation study using individual datasets while maintaining consistent pipeline parameters across four iterations. As shown in Table 11, VideoSAVi demonstrates dataset-agnostic improvements, achieving gains over the baseline InternVL2.5 model regardless of the source dataset. Star (Wu et al., 2021) videos particularly enhance complex reasoning benchmarks (NeXTQA +2.1 percentage points, EgoSchema +3.5), likely due to their rich situated contexts. Vidal (Zhu et al., 2024) contributes balanced improvements across benchmarks, while WebVid (Bain et al., 2021) exhibits stronger contributions to perception-oriented tasks (PerceptionTest +2.8). Notably, the performance gap between individual datasets and the combined approach (gap of 1.5-3.3 percentage points across benchmarks) demonstrates that VideoSAVi does not merely overfit to specific dataset preferences but rather benefits from diverse visual contexts. This finding reveals a critical insight: effective preference learning for video understanding requires exposure to varied temporal dynamics, visual compositions, and reasoning scenarios that single-source datasets cannot provide in isola-

| Dataset Source | Benchmark Performance | | | | | |
|---|---|---|---|---|---|---|
| | TempCompass | PerceptionTest | NeXTQA | MVBench | EgoSchema | LongVideoBench |
| *Baseline* | | | | | | |
| InternVL2.5 | 68.3 | 62.2 | 77.0 | 68.9 | 52.0 | 57.8 |
| *Single Dataset (1,000 videos)* | | | | | | |
| Star Only | $68.7_{+0.4}$ | $64.1_{+1.9}$ | $79.1_{+2.1}$ | $71.3_{+2.4}$ | $55.5_{+3.5}$ | $58.9_{+1.1}$ |
| Vidal Only | $68.6_{+0.3}$ | $64.5_{+2.3}$ | $78.5_{+1.5}$ | $70.2_{+1.3}$ | $54.6_{+2.6}$ | $58.5_{+0.7}$ |
| WebVid Only | $68.5_{+0.2}$ | $65.0_{+2.8}$ | $78.4_{+1.4}$ | $70.7_{+1.8}$ | $53.8_{+1.8}$ | $58.7_{+0.9}$ |
| **All Datasets** | $\mathbf{69.1}_{+0.8}$ | $\mathbf{66.1}_{+3.9}$ | $\mathbf{80.6}_{+3.6}$ | $\mathbf{74.0}_{+5.1}$ | $\mathbf{58.8}_{+6.8}$ | $\mathbf{59.8}_{+2.0}$ |

Table 11: **Dataset Ablation Analysis of VideoSAVi.** We evaluate the performance impact of using only a single source dataset (1,000 videos each) vs. the full combination across six benchmarks. Improvements over the baseline are shown as subscripts. While each individual dataset contributes to performance gains, their combination yields consistently superior results, demonstrating the importance of diverse video sources for comprehensive understanding. Star (Wu et al., 2021) videos particularly enhance reasoning benchmarks, Vidal (Zhu et al., 2024) shows balanced improvements, and WebVid (Bain et al., 2021) contributes strongly to perception tasks, but the integration of all sources creates the strongest overall results.

tion. The complementary nature of different video sources enables the model to develop more robust reasoning capabilities applicable across diverse benchmarks, confirming that dataset diversity serves as an implicit regularization factor in self-supervised preference optimization.

## A.4 DPO Training Dynamics

The DPO training dynamics in Figure 4 reveal critical insights into VideoSAVi's self-alignment process. The reward distribution exhibits a clear bifurcation pattern, with preferred responses achieving consistently higher rewards ($\mu \approx 1.5$) than dispreferred alternatives ($\mu \approx 0.5$). This substantial separation (average gap $\approx 1.0$) confirms the model's ability to differentiate quality even without external supervision. Gaussian smoothing ($\sigma = 15$) was applied to mitigate the inherent stochasticity in neural network training while preserving underlying trends. The rapid increase in classification accuracy (reaching $\approx 0.8$ by iteration 1000) demonstrates that the model quickly learns to distinguish between response qualities, with accuracy stabilizing despite continued training, indicating robust generalization rather than overfitting. Most notably, the reward gap's consistent growth throughout training suggests that the model continuously refines its understanding of quality differences rather than merely amplifying initial biases. The training loss profile exhibits three distinct phases: rapid early descent (iterations 0-500), steady intermediate refinement (iterations 500-1500), and convergence with minor oscillations (iterations 1500+). This pattern aligns with theoretical expectations for preference optimization and demonstrates that our self-alignment approach achieves parameter convergence without external reward signals. This validates the viability of purely self-supervised preference learning for improving video understanding.

## A.5 Generalization Test Bed Details

To rigorously evaluate the generalization capabilities of video understanding models, we construct a comprehensive test bed spanning multiple dimensions of generalization difficulty. This section details our methodology, data sources, and evaluation protocols.

### A.5.1 Dataset Composition

Our test bed comprises videos from diverse sources:

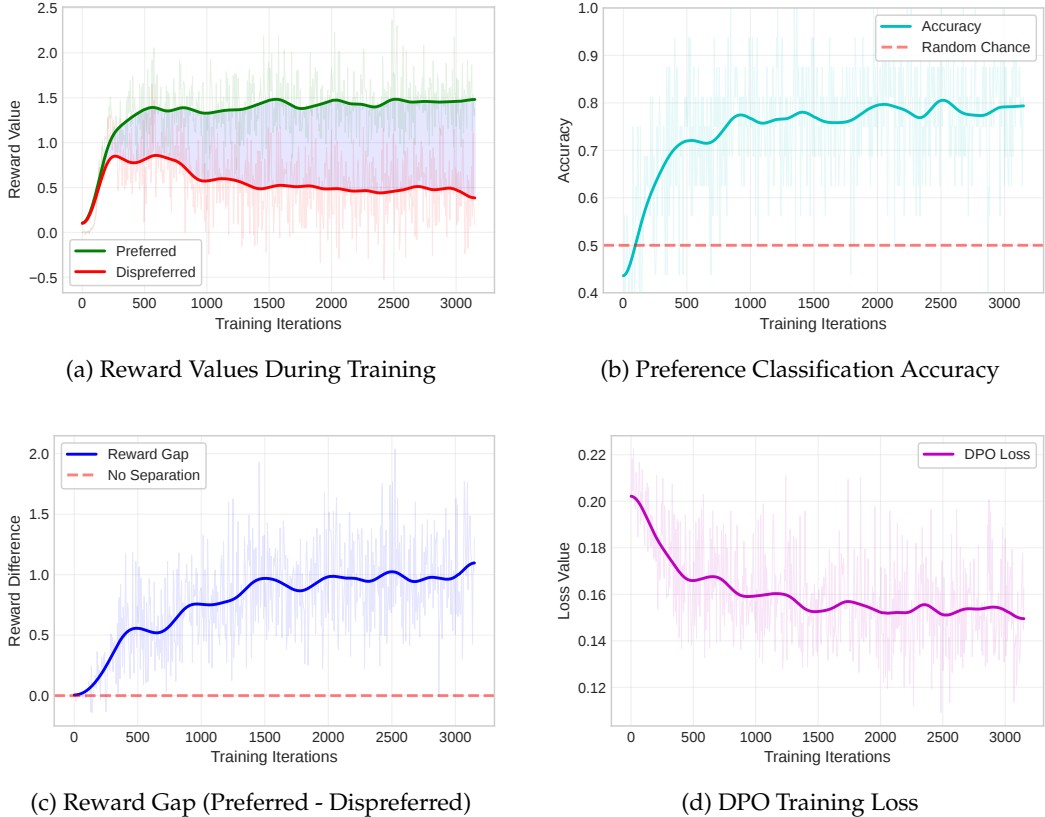

(a) Reward Values During Training

(b) Preference Classification Accuracy

(c) Reward Gap (Preferred - Dispreferred)

(d) DPO Training Loss

Figure 4: **Direct Preference Optimization training dynamics for VideoSAVi**. The visualizations show: (a) clear separation between preferred and dispreferred response rewards; (b) increasing preference classification accuracy that stabilizes at approximately 80%; (c) growing reward gap demonstrating effective preference learning; and (d) consistent loss reduction indicating stable convergence. These metrics confirm that our self-alignment approach successfully produces high-quality preference pairs that enable effective model optimization without external supervision.

1. **Training Distribution**: 100 videos each from Star (Wu et al., 2021), WebVid (Bain et al., 2021), and Vidal (Zhu et al., 2024) datasets (300 total), representing the model's original training domain.

2. **Cross-Video**: 150 additional unseen videos from the same datasets.

3. **Full OOD**: 150 videos from ActivityNet (Yu et al., 2019), representing a completely different domain with more complex activities.

For each video, we generate a rich set of questions using GPT-4o (Hurst et al., 2024) according to the prompt shown in Figure 13.

### A.5.2   Test Condition Categories

We design six distinct test conditions with progressively increasing difficulty:

1. **Training Distribution.** Standard format questions applied to training distribution videos. These serve as our baseline for measuring generalization gaps.

2. **Cross-Question.** Questions with complex syntax and uncommon vocabulary applied to *training distribution videos*. This tests linguistic generalization while keeping visual content consistent.

3. **Cross-Video.** Standard format questions applied to *unseen videos from the same datasets*. This tests visual generalization while keeping question patterns consistent.

4. **Full OOD.** Standard format questions applied to *ActivityNet videos*, testing both visual and domain generalization.

5. **Adversarial.** Deliberately challenging questions applied to *training distribution videos*, requiring focus on subtle or non-obvious elements.

6. **Compositional.** Multi-step reasoning questions applied to *training distribution videos*, requiring integration of multiple reasoning types.

### A.5.3 Evaluation Protocol

For each test condition, we evaluate models using multiple-choice accuracy, with each question accompanied by four possible answers (generated by GPT-4o). We maintain consistent evaluation protocols across all models to ensure fair comparison. To quantify generalization capability, we calculate (1) performance on each test condition, (2) the drop in performance from the training distribution to each other condition, and (3) the average generalization gap across all challenging conditions.

### A.5.4 Quality Check

To ensure the quality of our test bed, we conduct a verification step where we manually review a randomly sampled subset of 200 question-answer pairs across all conditions. The verification confirms that:

- 97.5% of questions are answerable from video content.
- 94.0% are categorized correctly according to their intended test condition.
- 98.0% have a single unambiguously correct answer.

This verification step validates the quality and reliability of our generalization test bed. The distribution of questions across different reasoning types (spatial, temporal, object-centric, action-centric) was balanced to avoid bias toward any particular reasoning capability.

### A.6 Qualitative Analysis

Figure 5 presents a detailed qualitative comparison between our VideoSAVi and the baseline InternVL2.5 (Chen et al., 2024) model across three distinct video understanding scenarios. Each example demonstrates how our proposed self-critique and preference optimization pipeline effectively addresses different types of reasoning errors, highlighting the improvements made in both temporal and spatial reasoning through our method.

In the first example (cooking scenario), the baseline model shows a common temporal hallucination by generating a non-existent microwave heating step. This represents a critical factual error in temporal reasoning where the model invents an action sequence not supported by visual evidence. VideoSAVi correctly identifies and generates an accurate representation of the action sequence, properly recognizing the transfer of the mixture between containers before dishwashing begins.

The second example (brick-making) highlights spatial reasoning deficiencies in the baseline model. The baseline model incorrectly positions the finished bricks in "vertical columns behind" the person and misidentifies the clay pit as being "to the right" of the work area. VideoSAVi produces a refined response that accurately captures the grid pattern arrangement to the left side of the person and the correct positioning of the clay pit directly in front of the person.

The third example (car cleaning) demonstrates object hallucination, where the baseline model incorrectly claims the person is using "soap and a brush" rather than identifying the pressure washer visible in the video frames. VideoSAVi not only correctly identifies the pressure washer but also accurately describes additional contextual elements like the outdoor setting and the organization of cleaning supplies.

The fourth example (dog competition) showcases VideoSAVi's ability to connect multiple events within a video. The baseline model hallucinates specific details about treats, judges, obstacle arrangement, and reward ribbons, none of which appear in the video. VideoSAVi produces a response that precisely captures the actual course elements (yellow-framed obstacles, jump bars, blue tunnel) and correctly identifies the spatial arrangement of spectators and tents.

These examples illustrate how our self-critique mechanism systematically identifies reasoning errors across both spatial and temporal dimensions, creating high-quality preference pairs that enable the model to learn from its own mistakes without external supervision. The consistent pattern of error correction across diverse scenarios demonstrates the generalizability of our approach, showing strong improvements in eliminating hallucinated objects, correcting spatial relationships, and accurately representing temporal sequences.

### A.7 Dataset Samples

Figure 6 showcases representative examples from our preference pair collection, illustrating how VideoSAVi effectively refines its reasoning capabilities. The examples reveal critical distinctions in temporal reasoning, where the model transitions from generating non-existent events (athlete celebrating) to precise temporal sequencing (immediate action following bar clearance) and from reversing causal sequences (blending after squeezing) to establishing correct procedural order. Similarly, for spatial reasoning, the preference signal guides the model from imprecise localization (beneath/partially covered) toward accurate spatial relationships (centrally positioned/surrounded) and from incorrect reference points (behind the leftmost cup) to precise positional understanding (underneath the middle cup). These preference pairs exemplify how our self-critiquing mechanism isolates specific reasoning failures without requiring human annotation or proprietary model distillation. The resulting preference signal contains subtle yet unambiguous distinctions that, when optimized through DPO, enable VideoSAVi to develop robust spatial-temporal reasoning capabilities grounded directly in visual evidence. This demonstrates the effectiveness of our approach in generating high-quality training signals through purely self-supervised means.

### A.8 Example of Self-Critiquing Mechanism for Improving Temporal and Spatial Reasoning

Figure 7 demonstrates VideoSAVi's self-critiquing mechanism across diverse video understanding scenarios. Our approach enables the model to identify and correct reasoning failures without external supervision. The examples shown are from the fourth iteration of self-training, where reasoning capabilities have been significantly enhanced through repeated preference optimization.

As illustrated in Figure 7, our self-critique pipeline systematically addresses both temporal and spatial reasoning errors. For temporal reasoning (first example), the critique identifies when responses are *too general* and lack *specific temporal details* about actions like the precise movements of a table tennis player during a rally. For spatial understanding (second example), the pipeline enhances detail about *relative positions* of subjects within scenes, such as specifying the exact seating position of a person in a green shirt. The hamburger drawing example (third) shows how the critique mechanism detects when a response *fails to mention* critical preparatory actions, while the food preparation example (fourth) highlights the correction of *significant spatial-temporal inaccuracies* in hand coordination.

By framing these critiques as preference pairs (in the next step), we align the model specifically to these dimensions while avoiding the introduction of biases that may arise when using external or proprietary models for critiquing. This approach mitigates failure modes that commonly affect video-LLMs: temporal hallucinations (generating non-existent sequences), spatial misrepresentations (incorrectly positioning objects), and detail omissions (missing critical visual evidence).

The advantage of our approach lies in maintaining end-to-end differentiability throughout the preference learning process. When the model identifies that a response is *too general* or

contains *significant inaccuracies*, it generates precise corrective signals focused on the specific reasoning dimension requiring improvement. Through iterative DPO refinement, these self-generated preference pairs enable VideoSAVi to progressively strengthen its reasoning across spatiotemporal dimensions without requiring external labels or feedback.

Unlike approaches that rely on external critique models, our self-alignment mechanism preserves internal video representations throughout the refinement process, avoiding the information loss that typically occurs when passing through different model architectures. This technical design choice enables more efficient learning from fewer examples, as demonstrated in the food preparation example where hand positioning and action synchronization are precisely captured.

### A.9 Prompt Design for Self-Aligning Video-LLMs

The effectiveness of VideoSAVi relies critically on carefully designed prompts that guide each stage of the self-alignment pipeline. Our prompt templates are written to elicit specific types of reasoning, generate high-quality self-critiques, and produce refined responses that address identified shortcomings without external supervision.

**Reasoning-Focused Question Generation.** Figure 8 presents our template for generating diverse and challenging questions that target specific reasoning capabilities. This prompt is structured to balance spatial and temporal understanding requirements with explicit instructions to focus on relationships that require careful attention. The spatial reasoning component emphasizes object positioning, visual details, and scene composition, while the temporal reasoning component focuses on event sequencing, cause-effect relationships, and state transitions. We deliberately avoid questions with trivial answers by instructing the model to prioritize aspects requiring deeper visual analysis. This prompt enables us to create an initial dataset of challenging questions and preference pairs, which are then used to train the model through DPO.

**Self-Critique Mechanism.** The core innovation of VideoSAVi lies in its ability to critique its own responses. Figure 9 shows our template for generating comprehensive assessments of reasoning errors. The prompt is carefully designed to evaluate four distinct aspects: spatial reasoning accuracy, temporal reasoning correctness, cross-modal consistency between claims and visual evidence, and overall response quality. For each identified issue, the model must specify the problematic statement, explain why it is incorrect, provide contradicting visual evidence, and assess the severity of the error. This structured approach ensures that critiques are specific and actionable rather than vague or general, enabling precise, targeted improvements in subsequent iterations.

**Response Refinement.** Figure 10 illustrates our template for generating improved responses through self-critique. This prompt guides the model to carefully balance preserving accurate information while correcting identified errors. We emphasize factual correctness, grounding in visual evidence, and logical consistency to prevent the introduction of new hallucinations during refinement. The prompt specifically targets factual errors, temporal ordering mistakes, spatial relationship misrepresentations, and reasoning flaws identified in the critique stage. This targeted refinement approach enables VideoSAVi to generate high-quality preference pairs that effectively isolate and improve specific reasoning capabilities.

**External Critique Generation.** For a fair comparison with external critique methods, we design the prompt shown in Figure 11. This template guides GPT-4o in analyzing response pairs to identify specific errors in dispreferred responses compared to preferred alternatives. The structured critique format highlights issues in spatial and temporal reasoning, factual inaccuracies, and missing critical details. This prompt serves as a benchmark for evaluating the effectiveness of VideoSAVi's self-critique approach against externally generated critiques.

**Error Analysis.** Figure 12 presents our error categorization template, used with GPT-4o to evaluate model improvements across iterations. This prompt defines various error types, including factual inaccuracies, temporal ordering mistakes, spatial misrepresentations, object hallucinations, causal reasoning failures, and detail omissions. By enforcing a structured output format, the template enables quantitative tracking of error reduction across iterative

refinements, providing objective metrics for VideoSAVi's self-improvement process. This experiment is essential in verifying that our approach genuinely corrects reasoning errors rather than reinforcing existing patterns.

**Generalization Assessment.** To rigorously evaluate generalization capabilities, we design the prompt in Figure 13. This template generates progressively more challenging question variants, including standard training distribution questions, complex cross-question variants, deliberately challenging adversarial questions, and multi-step compositional questions. By structuring the difficulty progression, we can measure VideoSAVi's generalization gap across different reasoning dimensions and compare it with alternative approaches. The significant reduction in the generalization gap (-5.2% for VideoSAVi vs. -16.3% for Hound-DPO) validates the effectiveness of our method in achieving robust generalization.

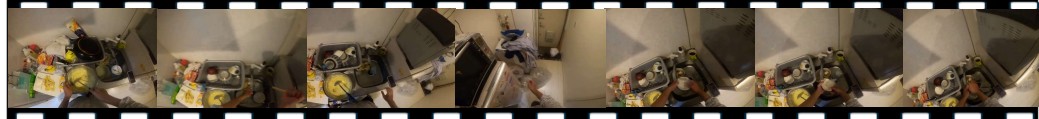

**Query:** What sequence of actions does the person perform with the yellow mixture before washing dishes?

**InternVL2.5:** The person stirs the yellow mixture, places it in the microwave to heat it, then returns to take it out before washing dishes in the sink. The heating step occurs between handling the mixture and dishwashing.

**VideoSAVi:** The person stirs the yellow mixture in a bowl, transfers some of it to another container, and then moves to wash dishes in the sink. The yellow mixture remains in the bowl until dishwashing begins.

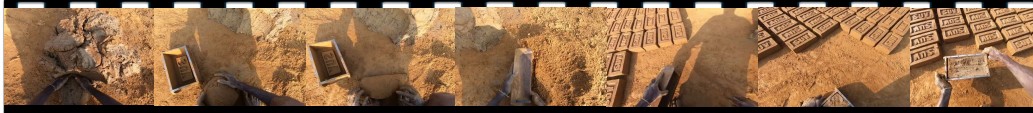

**Query:** How are the finished bricks arranged in relation to the person's work area and the clay pit?

**InternVL2.5:** The finished bricks are stacked in vertical columns behind the person. The clay pit is to the right of the work area, and the person moves between the pit and the brick arrangement area.

**VideoSAVi:** The finished bricks are arranged in a grid pattern to the left side of the person. They are positioned away from the clay pit, which is directly in front of the person.

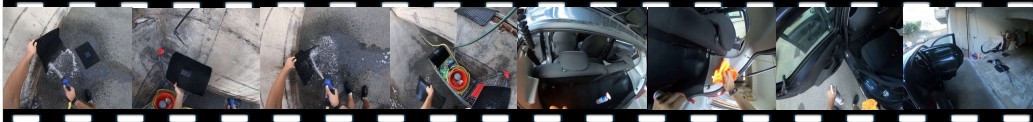

**Query:** What cleaning task is the person performing in this video?

**InternVL2.5:** The person is cleaning the car's floor mats using soap and a brush. The person scrubs each mat by hand after removing them from the vehicle.

**VideoSAVi:** The person is cleaning the car's floor mats with a pressure washer. They remove the black rubber mats from the vehicle, spray them with water in an outdoor area, and have their cleaning supplies organized in a shopping cart nearby.

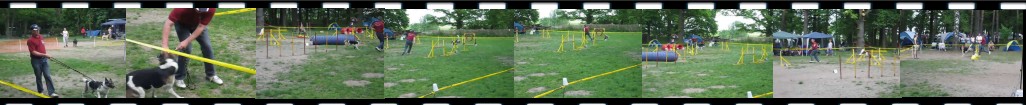

**Query:** Describe the video.

**InternVL2.5:** The video shows a dog agility competition where the person is using treats to guide the dog through the course. The obstacles are arranged randomly. There are five judges observing from different angles, and the event concludes with dogs receiving colored ribbons based on their performance.

**VideoSAVi:** The video shows a dog competition taking place in an outdoor park setting. There are yellow-framed obstacles forming a course, including jump bars and a blue tunnel. Multiple handlers guide dogs through the course while spectators watch from the sidelines where blue tents are set up. The black and white dog is seen navigating between obstacles with its handler.

Figure 5: **Qualitative examples demonstrating VideoSAVi's self-critique pipeline for improving video understanding**. Baseline model responses reveals reasoning errors: **hallucinated details** (judges, treats), **incorrect spatial relationships** (stacked columns, right side), and **fabricated details** (soap, scrubbing). Through our self-critiquing mechanism, VideoSAVi corrects these errors by: (1) generating accurate temporal sequences (yellow mixture handling), (2) precisely identifying spatial configurations (brick arrangement), and (3) eliminating object hallucinations (pressure washer vs. brush). Red text indicates errors in baseline responses and green text highlights accurate details in VideoSAVi responses.

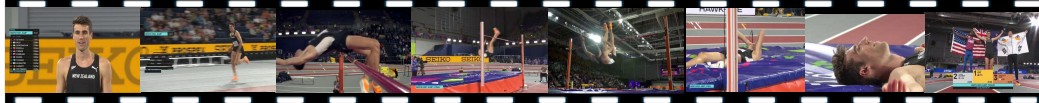

**Query:** Did the athlete lie down on the mat immediately after clearing the bar?

**Dispreferred Response:** The athlete first stood and celebrated before lying down on the mat.

**Preferred Response:** The athlete immediately lay down on the mat after clearing the bar.

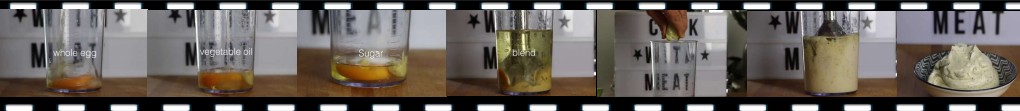

**Query:** Was the mixture blended before or after squeezing lime into the container?

**Dispreferred Response:** The mixture was blended after squeezing lime into the container.

**Preferred Response:** The mixture was blended before squeezing lime into the container.

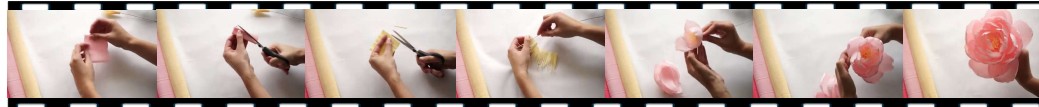

**Query:** How is the yellow material positioned relative to the pink petals in the final flower?

**Dispreferred Response:** The yellow material is positioned beneath, partially covered by the pink petals.

**Preferred Response:** The yellow material is positioned centrally, surrounded by the pink petals.

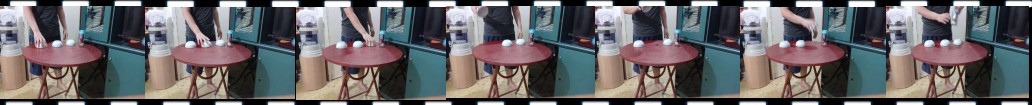

**Query:** Where does the person place the blue object relative to the white cups?

**Dispreferred Response:** The blue object is placed behind the leftmost white cup.

**Preferred Response:** The blue object is placed underneath the middle white cup.

Figure 6: **Examples of preference pairs from our dataset showing how VideoSAVi corrects reasoning errors through self-critiquing**. The top two examples demonstrate **temporal reasoning** improvements: accurately identifying action sequences (athlete immediately lying down) and event ordering (mixture blending before squeezing). The bottom two showcase **spatial reasoning** refinements: correctly representing positional relationships (yellow material centrally positioned versus underneath) and precise object localization (cup placement). These subtle yet critical distinctions form the training signal that enables VideoSAVi to learn fine-grained video understanding without external supervision. Red text indicates errors in dispreferred responses and green text highlights accurate details in preferred responses.

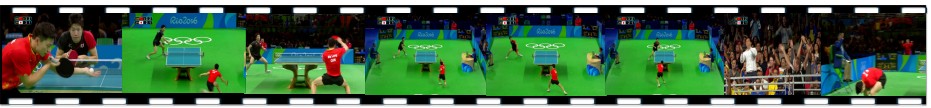

**Query:** What happens after the player in red serves?

**Initial Response:** After the red player serves, the opponent returns the ball and a brief rally occurs. The red player then makes a winning shot.

**Self-Critique:** The response is too general and lacks specific temporal details. It doesn't clearly indicate the exact movements of the red player during the rally.

**Refined Response:** After the red player serves, he steps back, shifts right, receives his opponent's return, and returns a forehand that his opponent fails to return properly, ending the rally.

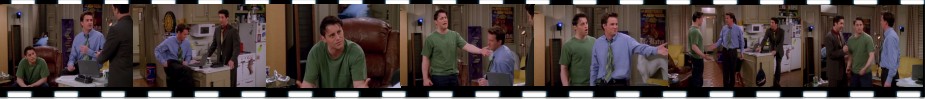

**Query:** How does the man in the green shirt's position change in the video?

**Initial Response:** The man in the green shirt is sitting on the leather couch and later he is standing in the middle of the room.

**Self-Critique:** The response needs more spatial details about his relative position to the other people and objects. The temporal sequence could be more clearly described.

**Refined Response:** Initially, the man in the green shirt is seated on the left side of the brown couch. Later, he has moved to a standing position in the center of the room with his right hand extended outward while facing the man in blue.

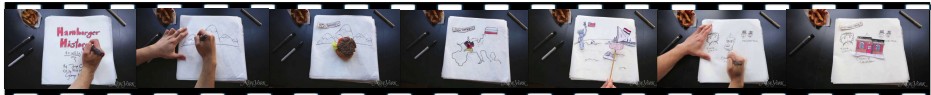

**Query:** What actions does the person take before placing the real hamburger on the drawing paper?

**Initial Response:** The person writes "Hamburger Master" at the top of the page before placing the real hamburger on the drawing paper. They are holding a pen and creating the title text as the initial step in their drawing process.

**Self-Critique:** The response is too general and lacks specific temporal details. It doesn't clearly identify all the observable actions that occur before the hamburger placement. The response fails to mention the exact positioning of the paper and how the person prepares the workspace.

**Refined Response:** Before placing the real hamburger on the paper, the person first positions the blank white paper, then writes "Hamburger Master" at the top of the page, completing the title. They hold the pen in their right hand while steadying the paper with their left hand.

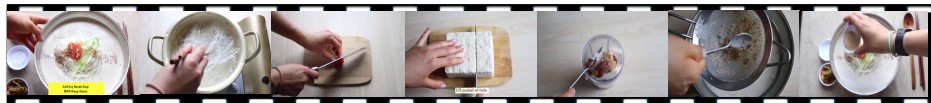

**Query:** How does the person coordinate their right and left hands for preparing the dish?

**Initial Response:** During the cutting process, the person uses their right hand to hold the knife while their left hand rests on the counter. They cut with the right hand in a downward motion while occasionally adjusting the food with both hands.

**Self-Critique:** The response contains significant spatial-temporal inaccuracies. It incorrectly claims the left hand rests on the counter during cutting, contradicting visual evidence. It fails to describe the specific positioning and movement of both hands in relation to each other and to the food.

**Refined Response:** The person begins by positioning the food on the cutting board using both hands. During the actual cutting, the left hand stabilizes the food item by pressing down from above, while simultaneously the right hand grips the knife for cutting. This coordinated hand positioning creates a spatial relationship where the left hand serves as support while the right hand performs the cutting action.

Figure 7: **Examples of VideoSAVi's self-critique pipeline for various video understanding tasks.** Red text indicates errors in initial responses, cyan text shows critique identified errors and green text highlights revised details in VideoSAVi responses.

**Reasoning-Focused Question Generation Prompt**

You are tasked with generating challenging and diverse questions about video content that test deep understanding of both spatial and temporal relationships. These questions should target specific reasoning capabilities that will help improve video understanding models.

**Spatial Reasoning Questions:** Generate questions that require:

- Identifying precise spatial relationships between objects/people in the scene.
- Detecting subtle visual details that might be easily overlooked.
- Comparing and contrasting different regions of the frame.
- Recognizing object attributes (color, size, shape, texture) and their spatial arrangement.
- Understanding occlusion, perspective, and relative positioning.
- Reasoning about visual composition and scene layout.

**Example spatial questions:**

- "What is the spatial relationship between the adult and child in the scene?"
- "How are the objects on the table arranged relative to each other?"
- "What visual details in the top-right corner distinguish it from the bottom-left?"
- "What is the configuration of people and furniture in the room?"

**Temporal Reasoning Questions:** Generate questions that require:

- Tracking the sequence and order of events.
- Understanding cause-effect relationships between actions.
- Identifying what happens before or after specific key moments.
- Reasoning about duration, speed, and timing of actions.
- Detecting transitions between states or scenes.
- Analyzing how entities change or move over time.

**Example temporal questions:**

- "What happens immediately before the person picks up the cup?"
- "What sequence of events leads to the child falling?"
- "How does the arrangement of objects change from the start to the end?"
- "What causes the dog to start running?"

**Guidelines for Question Generation:**

- Create questions that are answerable solely from the video content.
- Focus on aspects that require careful attention and reasoning.
- Avoid questions with trivial or immediately obvious answers.
- Ensure questions target both easy-to-observe and subtle elements.
- Generate questions that challenge understanding without requiring specialized knowledge.
- Balance questions across different areas of the frame and timepoints in the video.

For each video, generate 3 spatial reasoning questions and 3 temporal reasoning questions that probe different aspects of understanding.

Figure 8: Prompt template used to generate diverse spatial and temporal reasoning questions that target specific reasoning capabilities for self-alignment.

---

**Self-Critique Generation Prompt**

You are acting as a critical evaluator for a response to a video-based query. Your task is to meticulously analyze the response for errors, inconsistencies, and potential improvements.

Given the video content and the initial response, thoroughly examine the following aspects:

**Spatial Reasoning Assessment:** Analyze whether the response accurately represents spatial relationships visible in the video. Check if object positions are correctly described relative to one another. Verify that all significant visible objects are accounted for without hallucination. Examine if object attributes (color, size, shape) are accurately represented. Assess if spatial descriptions are precise and unambiguous.

**Temporal Reasoning Assessment:** Evaluate whether the sequence of events is correctly ordered. Check if cause-effect relationships between actions are logically sound. Verify that temporal markers (before, during, after) accurately reflect video content. Assess if the duration of actions or events is realistically represented. Examine if transitions between states or scenes are accurately described.

**Cross-Modal Consistency:** Analyze if claims made in the response are directly observable in the video. Identify any statements that contradict visual evidence. Check for speculative content not grounded in the video. Verify that the response addresses the specific query without tangential information.

**Response Quality:** Evaluate if the answer is appropriately comprehensive for the query. Check for logical contradictions within the response itself. Assess if the response maintains an appropriate level of detail.

For each identified issue, clearly specify:

1. The exact problematic statement or omission.

2. Why it is incorrect or problematic.

3. The relevant visual evidence from the video that contradicts or is missing.

4. The severity of the error (critical or minor).

Conclude your critique with a concise summary of the most significant issues that should be addressed in a refined response. Focus particularly on factual errors rather than stylistic concerns.

---

Figure 9: Prompt template for the critic model to generate comprehensive assessment of spatial, temporal, and logical errors in initial responses.

---

**Response Refinement Prompt**

You are tasked with refining a response to a video-based question based on a detailed critique. Your goal is to produce an improved answer that addresses all identified issues while maintaining accuracy.
Review the following materials carefully:

- The original question about the video.
- Your initial response to this question.
- A detailed critique identifying specific errors and issues.

**Refinement Guidelines:**

- Maintain factual correctness by ensuring all claims are directly observable in the video.
- Preserve useful and accurate information from the original response.
- Correct all errors and inconsistencies identified in the critique.
- Avoid introducing new speculation not supported by visual evidence.
- Ensure logical consistency throughout your refined response.
- Address the specific question directly and comprehensively.

**Important:** Focus on addressing the specific issues raised in the critique, particularly factual errors, temporal ordering mistakes, spatial relationship misrepresentations, or reasoning flaws.
Consider the provided critique of your previous response and provide an improved response that addresses these issues.

---

Figure 10: Prompt template for response refinement, which instructs the model to produce an improved answer that incorporates feedback from the self-critique while maintaining factual accuracy and logical consistency.

---

**External Critique Generation Prompt**

You are analyzing pairs of responses to video-based questions, where one response (Preferred) has been judged as superior to the other (Dispreferred). Your task is to generate a detailed critique of the Dispreferred response, identifying specific deficiencies compared to the Preferred response.

**Given Information:**

- Video.
- Question about the video.
- Preferred response (higher quality).
- Dispreferred response (lower quality).

**Critique Guidelines:** Generate a detailed critique of the Dispreferred response that:

- Identifies specific factual errors or inaccuracies.
- Highlights failures in spatial reasoning (object positions, relationships, attributes).
- Points out temporal inconsistencies (event ordering, causality, transitions).
- Notes missing details that appear in the Preferred response.
- Identifies logical flaws or contradictions.
- Analyzes how and why the Dispreferred response fails to address the question.

**Structure Your Critique:**

1. Summary of Core Deficiencies: Briefly summarize the main problems (1-2 sentences)
2. Spatial Reasoning Issues: Identify specific spatial errors or misunderstandings
3. Temporal Reasoning Issues: Highlight sequence/causality problems
4. Factual Errors: List specific incorrect claims or hallucinations
5. Comparative Analysis: Note key elements present in the Preferred response but missing or incorrect in the Dispreferred
6. Improvement Recommendations: Suggest specific changes to correct the identified issues

Be specific, precise, and reference exact details from both responses and the video. Focus on substantive issues rather than stylistic differences. Where possible, explain why certain errors would be particularly problematic for user understanding.

Figure 11: Prompt used with GPT-4o to generate external critiques of dispreferred responses from existing preference datasets.

---

**GPT-4o Evaluation Prompt for Error Analysis**

You are an expert evaluator assessing responses to video-based questions. Your task is to identify and categorize errors in the provided response.

**You will be given:**

1. A video.

2. A question about the video.

3. A model-generated response to evaluate.

**Evaluation Categories:** Analyze the response for the following error types:

**Factual Inaccuracy:** Statements that directly contradict what is visible in the video. Claims about objects, actions, or events that do not appear in the video.

**Temporal Ordering:** Incorrect sequencing of events (e.g., claiming A happened before B when the reverse is true). Misrepresentation of causal relationships between actions.

**Spatial Relationship:** Incorrect descriptions of object positions or arrangements. Misrepresentation of relative locations (left/right, above/below, etc.).

**Object Hallucination:** Mentioning objects or entities that are not present in the video. Attributing incorrect properties to objects that do exist.

**Causal Reasoning:** Incorrect inferences about why events occurred. Unsupported claims about motivations or intentions.

**Detail Omission:** Failing to mention critical elements necessary to answer the question. Overlooking important visual details that change the interpretation.

**Instructions:**

1. For each error you identify, specify the error type from the categories above.

2. Quote the problematic text from the response.

3. Explain why it's an error based on the video description.

4. Count the total number of errors in each category.

**Output Format:**

1. Error counts by category.

2. Brief examples of each error type found.

3. An overall assessment of response quality.

Be thorough in your analysis, as this evaluation will be used to measure model improvement across iterations.

---

Figure 12: Prompt used for GPT-4o to evaluate and categorize errors in model responses across iterations.

---

**Test Condition Question Generation Prompt for Generalization Assessment**

You will generate diverse question types to test a video understanding model's generalization capabilities across increasingly challenging conditions. For each video, create questions in the following categories:

**Training Distribution Questions:** Create standard format questions that follow typical patterns:

- "What object appears after the person sits down?"
- "What is the spatial relationship between [object A] and [object B]?"
- "How many people are visible in the scene?"

**Cross-Question Variants:** Reformulate standard questions using novel phrasing and more complex language:

- "Describe the causal relationship between the sitting action and subsequent object appearance"
- "Elaborate on the configuration of entities in relation to the central figure"
- "What sequential patterns of movement can be discerned among the visible actors?"

**Cross-Video Questions:** Apply standard question formats to unseen videos from the same datasets:

- Use the same question templates as Training Distribution.
- Test visual generalization with consistent question complexity.
- Focus on transferring learned patterns to new visual content.

**Adversarial Questions:** Create questions with deliberately misleading cues or that require focusing on non-obvious elements:

- "What is the main action occurring in the background?" (when foreground action is prominent)
- "What subtle change occurs to the leftmost object while attention would naturally focus on the center?"
- "Ignoring the primary movement, what secondary action follows the initial event?"

**Compositional Questions:** Formulate questions requiring multiple reasoning steps or integration of different reasoning types:

- "Compare the temporal order of actions on the left versus right sides of the frame"
- "How does the spatial configuration change from the beginning to the end of the sequence?"
- "What causal chain connects the initial object arrangement to the final state?"

For each video, generate 8 questions (2 per category) with 4 answer choices and 1 correct choice. Ensure questions are answerable from the video content, specific, and aligned with their respective category definitions. For adversarial questions, identify what makes them challenging (e.g., distraction, subtlety, misdirection).

Figure 13: Prompt used to generate diverse test conditions for generalization assessment. GPT-4o created questions across standard, cross-question, adversarial, and compositional categories to systematically evaluate VideoSAVi's ability to generalize beyond its training distribution.

