# OpenReview forum: "VideoSAVi: Self-Aligned Video Language Models without Human Supervision"
_colmweb.org/COLM/2025/Conference — COLM 2025_

### Official Review · Reviewer_nQcD · 2025-05-06

**Rating:** 8
**Confidence:** 3
**Ethics Flag:** 1

**Summary:**

The paper introduces VideoSAVi, a self-aligned video language model that improves video understanding without human supervision. It uses a self-critiquing mechanism to identify reasoning errors and generates preference pairs for training, achieving state-of-the-art performance on multiple benchmarks.

**Reasons To Accept:**

1. The proposed self-training framework with the quality-aware self-critiquing mechanism is novel and sound.

2. The proposed Direct Preference Optimization (DPO) enhances temporal and spatial reasoning for video understanding.

3. The experimental results confirm that VideoSAVi achieves state-of-the-art performance on several benchmark datasets.

3. The paper is well-written, easy to follow, and well-organized.

**Reasons To Reject:**

1. More state-of-the-art baseline models and benchmark datasets should be included to further confirm VideoSAVi's effectiveness.

---

> ### Author Response · Authors · 2025-06-02
>
> Thank you for your encouraging feedback. We value your recognition of our novel and sound self-training framework and the paper's clear writing.
>
> >More state-of-the-art baseline models and benchmark datasets should be included
>
> We have added two new state-of-the-art baseline models, (1) Qwen2.5VL-7B [3] and (2) Qwen2.5-Omni-7B [4] and applied VideoSAVi on these backbones. VideoSAVi delivers consistent gains on most benchmarks for Qwen2.5VL-7B and across **all** benchmarks for Qwen2.5-Omni-7B. Since, these models already underwent post-training refinements using DPO, the absolute gains are lower compared to other VLM backbones.
>
> | Model | Perception | NeXTQA | MVBench | EgoSchema | LongVideoBench | VideoMME | Vinoground | VSI-Bench |
> |-------|----------------|--------|---------|-----------|----------------|----------|------------|-----------|
> | Qwen2.5VL | 68.6 | 75.8 | 65.2 | 60.9 | 60.7 | 62.2 | 27.9 | 36.0 |
> | +VideoSAVi | **68.9** | **75.9** | **65.4** | **61.0** | 60.7 | **62.5** | **28.0** | **36.4** |
> | Qwen2.5-Omni | 66.6 | 75.2 | 63.0 | 61.8 | 56.9 | 58.8 | 24.1 | 22.7 |
> | +VideoSAVi | **66.9** | **75.6** | **64.3** | **61.9** | **57.2** | **59.0** | **24.2** | **23.0** |
>
>
> We also provide evaluation on three additional benchmarks: VideoMME [5] (long video reasoning), Vinoground [6] (temporal reasoning), and VSI-Bench [7] (spatial reasoning). **VideoSAVi achieves a new SOTA on VSI-Bench (38.9%) in the 7B model category** and competitive performance across all benchmarks.
>
> | Model | VideoMME | Vinoground | VSI-Bench |
> |-------|----------|------------|-----------|
> | **(1) Baseline Models** |
> | InternVL2.5 | 63.1 | 29.4 | 35.8 |
> | + SFT | 63.2 | 29.2 | 35.0 |
> | + Hound-DPO | 60.7 | 28.6 | 34.7 |
> | + TPO | **63.8** | 29.1 | 34.8 |
> | **(2) State-of-the-Art Models** |
> | LLaVA-NeXT-Interleave | 48.3 | 15.2 | 29.3 |
> | Qwen2-VL | 55.3 | 24.8 | 31.4 |
> | LLaVA-OneVision | 58.2 | 25.0 | 35.0 |
> | LLaVA-Video | 62.4 | 24.3 | 35.7 |
> | Qwen2.5VL-7B | 62.2 | 27.9 | 36.0 |
> | Qwen2.5-Omni-7B | 58.8 | 24.1 | 22.7 |
> | **(3) Preference-Optimized Models** |
> | LLaVA-Hound-DPO | 34.2 | 23.8 | - |
> |I-SRT | 34.7 | 23.7 | - |
> | LLaVA-Video-TPO | 62.4 | 24.0 | 35.9 |
> | VideoSAVi | $\underline{63.6}$ | **29.6** | **38.9** |
>
>
> ---
> [3] Bai, Shuai, et al. "Qwen2. 5-vl technical report." arXiv preprint arXiv:2502.13923 (Feb 2025).
>
> [4] Xu, Jin, et al. "Qwen2. 5-omni technical report." arXiv preprint arXiv:2503.20215 (March 2025).
>
> [5] Fu, Chaoyou, et al. "Video-mme: The first-ever comprehensive evaluation benchmark of multi-modal llms in video analysis." arXiv preprint arXiv:2405.21075 (2024).
>
> [6] Zhang, Jianrui, Mu Cai, and Yong Jae Lee. "Vinoground: Scrutinizing LMMs over Dense Temporal Reasoning with Short Videos." arXiv preprint arXiv:2410.02763 (2024).
>
> [7] Yang, Jihan, et al. "Thinking in space: How multimodal large language models see, remember, and recall spaces." arXiv preprint arXiv:2412.14171 (2024).

---

> > ### Comment · Reviewer_nQcD · 2025-06-10
> >
> > Thanks for the rebuttal. I will keep my score.

---

> ### Author Response · Authors · 2025-06-07
>
> Dear Reviewer, Thank you again for your valuable feedback. As the discussion deadline is June 10, we wanted to ensure that we have addressed all your questions. We welcome any additional thoughts you may have.

---

### Official Review · Reviewer_XAqh · 2025-05-11

**Rating:** 6
**Confidence:** 4
**Ethics Flag:** 1

**Summary:**

This work proposes VideoSAVi, a self-training video understanding pipeline that reasons over the model’s initial responses to generate improved ones, creating preference pairs. These pairs are then used for DPO training, enhancing the model’s performance without human supervision.

**Questions To Authors:**

Have you explored directly applying critique and improvement prompts to the model (without preference pair construction and training) to optimize benchmark performance? How do the results compare?

**Reasons To Accept:**

- Significant improvements are demonstrated across four video QA benchmarks, with an average gain of +3.4 points over six benchmarks.
- The memorization and generalization experiments (Table 3) indicate robustness in OOD cases, validating the method’s generalizability.
- The reward value study (Figure 4) shows that self-constructed preference pairs stabilize DPO training, avoiding collapse.

**Reasons To Reject:**

While the experiments are comprehensive, some evaluation settings lack rigor, raising concerns:

- Insufficient baseline details: Missing key information on data sources, scalability, and training setups, which are crucial for fair comparison.
- Limited data scaling in analysis experiments: While the main experiments use 4,000 videos and 24,000 preference pairs, some ablation studies and appendix analyses rely on only 100–300 videos, weakening their reliability.
- Ill-formed analysis in Table 5: GPT-4o is used as a critique model, but its own performance is not reported, making it difficult to assess the critique prompt’s quality across models.

---

> ### Author Response · Authors · 2025-06-02
>
> We sincerely thank the reviewer for their detailed feedback and highlighting the significant improvements across benchmarks.
>
> >Insufficient baseline details
>
> We detail VideoSAVi’s data sources, training setup, and hyperparams in Sec. 4 (lines 217–224), and all prompts in Appendix A.9. For a fair comparison with other preference-optimized methods (Hound-DPO and TPO), we applied their publicly available preference data directly to our InternVL2.5 backbone using the same setup as VideoSAVi.
>
> Additional results below (expanded Table 4) show consistent performance improvements across multiple backbones, confirming the **model-agnostic** and generalizable nature of our approach.
>
> | Model                               | Perception | NeXTQA | MVBench | EgoSchema | LongVideoBench |
> | :---------------------------------- | :-------------: | :----: | :-----: | :-------: | :------------: |
> | LLaVA-OneVision                     |      57.5       |  79.3  |  56.7   |   64.0    |     56.3       |
> |    +Hound-DPO                     |        55.8        |   78.1    |    55.3    |     62.0     |       54.8        |
> |    +TPO                           |        58.4        |   80.6    |    57.9    |     **65.8**     |       57.5        |
> |    +VideoSAVi                     |    **60.8**     | **80.9** | **59.9** | 65.7 |   **58.7**     |
> | | | | | | |
> | Qwen2-VL                            |      62.1       |  75.6  |  64.9   |   59.2    |     55.6       |
> |    +Hound-DPO                     |        62.3        |   75.5    |    65.1   |     59.8     |       55.4        |
> |    +TPO                           |        63.0        |   75.9    |    66.4    |     61.7     |       56.0        |
> |    +VideoSAVi                     |    **65.4**     | **78.8** | **68.3** | **62.8** |   **58.1**     |
>
> ---
> >Limited data scaling in analysis experiments
>
> We scaled up our ablation studies for stronger validation. Findings remain unchanged, confirming VideoSAVi's effectiveness in reducing errors and improving generalization.
>
> 1.  **Reasoning Errors Across Iterations (Table 2):** Increased the eval set from 200 to **1200 videos** to more robustly show error reduction (78.4% decrease).
>
>     | Error Type             | Iter1 | Iter2 | Iter3 | Iter4 |
>     | :--------------------- | :---: | :---: | :---: | :---: |
>     | Factual Inaccuracy     |  287  |  198  |  127  |   73  |
>     | Temporal Ordering      |  243  |  162  |  108  |   52  |
>     | Spatial Relationship   |  208  |  147  |   92  |   48  |
>     | Object Hallucination   |  169  |  118  |   71  |   33  |
>     | Causal Reasoning       |  132  |   91  |   58  |   27  |
>     | Detail Omission        |  118  |   87  |   44  |   16  |
>     | **Total**              | 1157 | 803 | 500 | **249** |
>
> 2.  **Memorization vs. Generalization (Table 3):** Expanded our test bed (Appendix A.5) from 240 to **600 videos**, matching our main benchmarks. VideoSAVi still shows the smallest generalization gap.
>
>     | Test Condition         | VideoSAVi | TPO   | H-DPO |
>     | :--------------------- | :-------: | :---: | :---: |
>     | Training Dist.         |   74.5    | 71.2  | 69.8  |
>     | Cross-Question         |   70.2    | 62.5  | 56.2  |
>     | Cross-Video            |   69.8    | 61.8  | 54.1  |
>     | Full OOD               |   68.7    | 59.4  | 51.3  |
>     | Adversarial            |   67.9    | 57.8  | 48.5  |
>     | Compositional          |   68.3    | 58.6  | 49.2  |
>     | **Avg. Gen. Gap**      | **-6.4**  | -13.5 | -17.9 |
>
> ---
>
> >Ill-formed analysis in Table 5
>
>
> GPT-4o outperforms 7B models on most benchmarks:
>
> | Benchmark           | GPT-4o |
> | :------------------ | :----------: |
> | TempCompass         |     73.8     |
> | Perception Test     |     72.1     |
> | NeXTQA              |     88.1     |
> | MVBench             |     64.6     |
> | EgoSchema           |     66.2     |
> | LongVideoBench      |     66.7     |
>
>
> However, VideoSAVi’s fully self-contained critique outperforms external critiques like GPT-4o (as shown in **Table 5**), by keeping preference pairs closely aligned with the model’s own output distribution throughout iterative refinement. While GPT-4o doesn't generate strictly off-policy data, it can nudge responses into lower-probability regions, making DPO less effective due to the KL-divergence constraint. In contrast, VideoSAVi stays **on-policy**, enabling more stable and effective DPO learning.
>
> ---
> >Have you explored directly applying critique and improvement prompts?
>
> Critiques identify errors for each video–question–response instance, but these insights can't be directly used to improve Video-LLMs across different video understanding tasks. Instead, we use the error insights to construct preference pairs and apply DPO to train the model to generate more accurate responses.
>
> We tested Chain of Thought prompting with the baseline model. It yielded marginal gains on most benchmarks and significantly reduced performance on NeXT-QA. Please see our response to Reviewer CBX8 for details.

---

> > ### Comment · Reviewer_XAqh · 2025-06-03
> >
> > Thank you for your response. I have no more questions and would like to raise my score.

---

> > > ### Author Response · Authors · 2025-06-03
> > >
> > > Thank you. We are glad our response addressed your concerns and will incorporate your feedback into the revised paper.

---

### Official Review · Reviewer_CBX8 · 2025-05-12

**Rating:** 7
**Confidence:** 5
**Ethics Flag:** 1

**Summary:**

This paper proposes a framework to align video large language models (video-LLM) with human preference without human supervision. Using a self-critiquing mechanism, the framework gets feedback from other AIs and refine the response. Then, fine-tune/align video-LLM with the refined response using a standard post-training method direct preference optimization (DPO). The proposed method achieves strong performance on various video benchmarks, including MVBench, EgoSchema, and PerceptionTest.

**Questions To Authors:**

- **Side effect/overfitting of self-training.** The idea of self-training idea is intriguing but as with other post-training/fine-tuning methods, it may negatively impact abilities originally learnt during pre-training. Did you evaluate the performance of the self-trained models on the tasks that they previously handle well?
- **Computational cost of experiments.** In the experiment section, the experiments are conducted on two GPUs (NVIDIA L40S 48GB). How many GPU hours were required to conduct your experiments?
- Key difference between self-training and chain-of-thought?

**Reasons To Accept:**

* **Emerging topic.** Self-training is an emerging approach that can improve the performance of multimodal large language models without the need for additional annotations.

* **Strong empirical results.** The proposed method achieves state-of-the-art performance and demonstrates significant improvement over vanilla LLMs. Its effectiveness is validated across various benchmarks, including MVBench, NeXTQA, PerceptionTest, and EgoSchema.

* **Robustness analysis.** The robustness of the proposed method is analyzed in Table 3, demonstrating that under challenging test conditions, it exhibits the smallest performance degradation compared to baseline methods.

**Reasons To Reject:**

* (minor) **Overclaim.** The authors claim that self-training is more effective than increasing model parameters when discussing the results in Table 6. However, except for two cases with marginal gains 0.3 and 0.4, in most cases larger models achieved better performance. “These findings confirm that addressing … greater performance gains than merely increasing model capacity” should be toned down.

* (minor) **Manual efforts and comparison with longer prompts.** In the appendix, the authors provide actual self-critique generation prompt. The proposed method requires implicit human supervision. The human manually designed the prompts while monitoring the quality of responses. Also, as existing methods, manual/semi automated filter seems needed for DPO with refined responses. So, in some sense, this pipeline also requires human supervision. Self-training is good but the training with refined responses may not be perfect. Instead, using the criteria, chain-of-thoughts or conversation style generation is possible. With long, elaborated prompts through multiple iterations (CoT/conversation w/ good prompts), it might achieve similar or even better performance  than the proposed method. Do you have analysis regarding this?

---

> ### Author Response · Authors · 2025-06-02
>
> Thank you for your feedback. We appreciate your acknowledgment of our strong empirical results and robustness.
>
> > Overclaim
>
> Thank you for this observation. We agree and will revise Table 6 to clarify that VideoSAVi offers **complementary benefits** to parameter scaling as a **cost-effective alternative**, while acknowledging that larger models generally achieve better absolute performance.
>
> ---
> >Manual efforts
>
> The initial prompt design requires a one-time human effort. Beyond that, the VideoSAVi pipeline is fully automated. The monitoring you mentioned refers to our post-training analysis, which was done solely for evaluation purposes.
>
> We agree that developing automated methods to filter noisy preferences is a valuable direction and we are actively exploring it. In the meantime, we ran a new experiment showing that VideoSAVi is highly robust to noisy preferences. Please refer to our response to Reviewer vG49 for more details.
>
>
> >Comparison with longer prompts (e.g., CoT)
>
> To investigate your hypothesis, we conducted an experiment by appending a CoT instruction to the baseline model. The table below shows that CoT prompting resulted in only marginal improvements on most benchmarks and significantly reduced performance on NeXT-QA. In contrast, VideoSAVi achieved higher performance gains across all benchmarks. For this experiment, we increased the output token limit to 4096. The prompt used is provided below.
>
>
> | Model| Perception | NeXTQA | MVBench | EgoSchema | LongVideoBench |
> | :------------------------------ | :-------------: | :----: | :-----: | :-------: | :------------: |
> | InternVL2.5          |      62.2       |  77.0  |  69.8   |   52.0    |      57.8
> |  +CoT           |         62.3         |   75.8 |  69.9   | 52.6      |  58.0                     |
> |+VideoSAVi               |      **66.1**       |  **80.6**  |  **74.0**   |   **58.8**    |      **59.8**      |
>
>
> ```python
> prompt = """Before providing your final answer, please follow these steps:
> 1. Carefully analyze the video frames and the question. Identify and briefly describe the key visual elements or temporal events in the video that are most relevant to answering this question.
> 2. Explain your reasoning process step-by-step. How do you connect the visual information to the question?
> 3. Evaluate each option against your reasoning and the visual evidence. Explain why you are choosing one option and eliminating others.
> 4. Finally, state your answer clearly. If it's multiple choice, use the format "The final answer is (X)"."""
> ```
> ---
> >Side effect/overfitting of self-training
>
> We ran additional experiments to evaluate whether VideoSAVi preserves the core capabilities of its InternVL2.5 backbone across diverse benchmarks. As noted in InternVL2.5's technical report [2], these benchmarks/capabilities were part of the model's pre-training stage. The results show that VideoSAVi largely maintains or slightly improves performance across benchmarks.
>
> | Ability Assessed          | Benchmark      | InternVL2.5 | InternVL2.5+ VideoSAVi |
> |:------------------------- |:-------------- |:----------------------:|:-----------------:|
> | Multimodal Understanding  | MMMU (val)     |          53.1          |     **53.2**      |
> | Document Understanding    | DocVQA (val)   |          91.3          |     **91.9**      |
> | OCR / Text-based VQA      | TextVQA (val)  |          79.2          |     **79.8**      |
> | Pure Language (Reasoning) | GSM8k (0-shot) |        **77.4**        |       76.9        |
> | Video Captioning          | VideoChatGPTBench|        2.73          |    **3.0**
> | Image Hallucination       | POPE (Avg)     |          90.1          |     **90.3**      |
>
> [2] Chen, Zhe, et al. "Expanding performance boundaries of open-source multimodal models with model, data, and test-time scaling." arXiv preprint arXiv:2412.05271 (2024).
>
> ---
> >Computational cost of experiments.
>
> For VideoSAVi's 4-iteration pipeline on InternVL2.5, the approximate costs were as follows:
> * **Preference Generation** took 12h on 2 NVIDIA L40S GPUs (24 GPU hours per iteration).
> * **DPO Training** also took 12h on 2 GPUs (another 24 GPU hours per iteration).
>
> In total, the complete pipeline required (24 + 24) × 4 = **~192 GPU hours.**
>
> ---
> >Key difference between self-training and chain-of-thought?
>
> The key difference between **self-training** and **chain-of-thought** lies in **when** and **how** they improve model behavior:
> * **Self-training** (e.g., VideoSAVi) improves the model **during training** by modifying its parameters using supervised signals, such as preference pairs from self-generated data. The goal is to enhance the model’s general reasoning capabilities **permanently**.
> * **Chain-of-thought** is an **inference-time** technique. It improves reasoning for individual queries by prompting the model to break down its thinking into intermediate steps. However, it does **not change the model’s parameters** or have lasting effects.

---

> > ### Comment · Reviewer_CBX8 · 2025-06-03
> >
> > I appreciated the authors for the elaborated response, including additional results. All my concerns have been addressed by the rebuttal and it's a bit surprising that no overfitting has occurred. Could do you have any intuition or components in your method that prevents the model from overfitting/forgetting?

---

> > > ### Author Response · Authors · 2025-06-03
> > >
> > > Thank you for the positive feedback. We address overfitting through several key design choices:
> > >
> > > **Frozen Vision Components + LoRA**: We freeze the vision encoder and projector of the baseline model and apply low-rank LoRA adapters (rank 8) to the LLM, updating only 0.1% of its parameters. This preserves the original visual feature extraction and focuses improvements on **vision-text alignment**.
> > >
> > > **DPO regularization**: The KL-divergence constraint in DPO regularizes learning by limiting divergence from the reference policy, helping to prevent overfitting.
> > >
> > > **On-policy preference generation**: Our self-generated preference pairs stay within the model's natural output distribution, avoiding the instability that can occur with external preference data.
> > >
> > > Because we do not alter the visual backbone, the model retains its visual understanding. Instead, VideoSAVi **enhances how the language model interprets and reasons over existing visual features**.

---

### Official Review · Reviewer_vG49 · 2025-05-13

**Rating:** 6
**Confidence:** 4
**Ethics Flag:** 1

**Summary:**

### Quality

* The manuscript is well‑written.
* Tables and figures clearly illustrate the benefits of the proposed method.

### Clarity

* The presentation flows logically and is easy to follow.
* All necessary technical details are provided.

### Originality

* The approach is moderately original.
* Self‑training / thought‑refinement to replace human annotation or external models has been studied extensively, but its application to **video understanding** remains relatively under‑explored.

### Significance

* The method shows consistent, model‑agnostic gains over baselines on several video benchmarks.
* However, it is **not** compared with the latest SOTA on some datasets—for example, *Video Instruction Tuning with Synthetic Data* (LLaVA‑Video‑7B, Oct 2024) scores **83.2** on NEXT‑QA and **67.9** on PerceptionTest, outperforming the results reported here.

---

**Overall:** The paper is clear and of good quality, with modest originality and solid—though not state‑of‑the‑art—significance.

**Questions To Authors:**

1. Do you have plans to report results that **directly compare** your method with the current SOTA **LLaVA‑Video‑7B** on NEXT‑QA and PerceptionTest?
2. What automated strategy can you adopt to **detect or filter hallucinated revisions at scale** to ensure the reliability of the DPO training signal?
3. Did you perform **decontamination** of the videos used for generating training data (e.g., removing overlaps with benchmark datasets)? If so, how was this done?
4. Have you evaluated the approach on **weaker video‑LLMs**—particularly those that follow instructions poorly or hallucinate more? Does the self‑training help or harm such models, and is there a performance threshold where it becomes detrimental?

**Reasons To Accept:**

1. **Timely exploration of self‑training for video LLMs**

   * Self‑training has been richly studied for text‑only LLMs, but **video understanding remains under‑explored**.
   * The authors’ method outperforms established alignment-based baselines such as **TPO** and **HOUND‑DPO**, demonstrating that a carefully designed, *self‑critique* loop can yield larger gains.
   * This work therefore advances the frontier of *video* self‑training and provides a concrete recipe others can build on.

2. **Consistent, model‑agnostic improvements on multiple benchmarks**

   * On **InternVL 2.5**, the proposed framework delivers a non‑negligible improvement of

     $$
       \Delta = \text{+}3.6 \text{ points (NEXT‑QA)}, \quad
       \text{+}6.8 \text{ points (EgoSchema)}
     $$

     over the baseline (exact numbers in Table 2). And it get new SOTA on MVBench with 74.0.
   * Similar gains transfer to other Video-LLMs, including **Video‑LLaVA** and **LLaVA‑OneVision**, indicating the approach is *model‑agnostic* and not tailored to a single architecture.
   * The breadth of benchmarks and backbones constitutes a **comprehensive experimental setup** that convincingly demonstrates robustness.

3. **Clear and accessible presentation**

   * The paper is *well‑written* and logically structured; key details (algorithm, training schedule, hyper‑parameters) are easy to locate.
   * Figures and tables neatly illustrate the self‑training pipeline and quantitative gains, making the contribution **simple yet effective** for readers to reproduce.

*In sum, the work offers a timely and rigorous exploration of self‑training for video LLMs, shows consistent cross‑model improvements, and is presented with exemplary clarity.

**Reasons To Reject:**

1. **Missing comparison with current SOTA**

   * The recent baseline *Video Instruction Tuning with Synthetic Data*—**LLaVA‑Video‑7B** (Oct 2024)—achieves

     $$
       83.2 \text{ on NEXT‑QA}, \qquad 67.9 \text{ on PerceptionTest},
     $$

     clearly outperforming all results reported in this paper.
   * Omitting such a strong, publicly available model prevents readers from gauging true progress and risks overstating the contribution.

2. **No scalable mechanism to prevent hallucinations in self‑training pairs**

   * The pipeline relies on the **same \~7 B base model**—with different prompts—to

     1. generate questions,
     2. draft answers,
     3. self‑critique, and
     4. produce revised answers.
   * These *(initial ✕ revised)* pairs are fed directly into a DPO stage. Although a human study over 516 videos finds **≈30 %** of the pairs are low‑quality, this filter is *not* applied to the full training set, leaving substantial noise unchecked.
   * Small video‑LLMs are prone to hallucinating fine‑grained visual details; without automatic verification, the preferred labels may be unreliable and could degrade rather than improve the model.

> **Overall:** By omitting the current SOTA and lacking a scalable quality‑control mechanism, the paper’s empirical claims remain unconvincing.

---

---

> ### Author Response · Authors · 2025-06-02
>
> We appreciate your thorough review and recognizing our timely exploration of self-training for video-LLLMs.
>
> > Missing comparison with current SOTA (e.g. LLaVA‑Video‑7B)
>
> **LLaVA-Video-7B is included in Table 1**. We show that VideoSAVi outperforms LLaVA-Video on 5 out of 6 benchmarks, based on our reproduced scores using lmms-eval (32 frames). Additionally, the LLaVA-Video authors acknowledge benchmark contamination, noting that their training data includes NeXT-QA and Perception Test, making direct comparison difficult.
>
>
> ---
> >No scalable mechanism to prevent hallucinations ... The pipeline relies on the same ~7B base model to generate questions, self-critique ...
>
> VideoSAVi’s pipeline is iterative: while Iteration 1 uses the initial 7B base model, each subsequent iteration builds on the preference-optimized model from the previous stage. This self-improvement is evidenced by a 79.5% reduction in reasoning errors from Iteration 1 to 4 (Table 2) and consistent benchmark gains across iterations (Appendix A.1, Fig. 3).
>
> ---
> >These (initial ✕ revised) pairs are fed directly into a DPO stage ... w/o removing noisy preferences
>
> To test the robustness of VideoSAVi to noisy preferences, we conducted a new experiment where we flipped labels for 30% of 24,000 preference pairs to introduce incorrect signals, then ran DPO. Despite the 30% noise, performance still exceeded the baseline on all benchmarks, indicating that the remaining preferences provided a sufficiently strong learning signal. Naturally, higher-quality preferences yield even better results.
>
> |Model|Perception|NeXTQA|MVBench|EgoSchema|LongVideoBench|
> | :-------------------------- | :-------------: | :----: | :-----: | :-------: | :------------: |
> | InternVL2.5      |      62.2       |  77.0  |  69.8   |   52.0    |     57.8       |
> | +VideoSAVi w\ Noisy Prefs |        64.9        |   77.8    |    71.2    |     57.1     |     58.2 |
> | +VideoSAVi      |      **66.1**       |  **80.6**  |  **74.0**   |   **58.8**    |     **59.8**       |
>
> ---
> >Small video‑LLMs are prone to hallucinations ... they could degrade rather than improve.
>
> As shown in **Table 6**, we applied the full VideoSAVi pipeline to the 1B and 2B versions of InternVL, both of which outperformed the baselines across all benchmarks. Below we show *successful* application of VideoSAVi to two additional 3B backbones:
> | Model             | Perception| NeXTQA | MVBench | EgoSchema | LongVideoBench |
> | :---------------- | :-------------: | :----: | :-----: | :-------: | :------------: |
> | Qwen2.5VL-3B      |       63.1         |  74.9    |    63.5    |   53.6      |     **55.8**         |
> |    + VideoSAVi   |       **63.7**         |   **75.0**    |   **63.7**    |     **54.1**     |       55.7        |
> | | | | | | |
> | Qwen2.5-Omni-3B    |     65.2         |   69.1    |    62.6    |     56.2     |       53.8        |
> |    + VideoSAVi   |       **65.4**         |   **69.7**    |    **63.8**    |     **56.5**     |       **54.1**        |
>
> ---
> >What automated strategy can you adopt to detect or filter hallucinated revisions at scale?
>
> Thank you for the suggestion. We agree that this is an important direction and hope to address it more thoroughly in future work. In the meantime, to address your question, we applied an automated method from recent literature [1]. Using our VideoSAVi model as a preference classifier, we assigned scores to generated responses and performed DPO using only high and low scoring responses (i.e. preference pairs). Results, shown below, indicate that automated filtering yields minimal benefit and self-generated pairs are already of sufficient quality.
>
> | Model                                     | Perception | NeXTQA | MVBench | EgoSchema | LongVideoBench |
> | :---------------------------------------- | :-------------: | :----: | :-----: | :-------: | :------------: |
> | VideoSAVi             |      **66.1**       |  80.6  |  **74.0**   |   58.8    |     59.8       |
> | VideoSAVi + Critique Score Filtered |       66.0         |   **80.7**    |    73.2    |     **59.1**     |       **60.1**        |
>
> [1] Garg, Shivank, et al. "IPO: Your Language Model is Secretly a Preference Classifier." arXiv preprint arXiv:2502.16182 (2025).
>
> ---
> > Did you perform decontamination of the videos used for generating training data.
>
> Yes, we ensured training data decontamination by cross-checking our training sources (Star, Vidal, WebVid) against the evaluation benchmarks to avoid any overlap. Specifically, we identified and removed videos from the Star dataset that also appear in MVBench.
>
> ---
> >Have you evaluated the approach on weaker video‑LLMs?....is there a performance threshold?
>
> Yes, **Table 4** presents results for two early video LLMs released in early 2024: (1) VideoLLaVA and (2) LLaVA-NeXT-Interleave. While we observe no detrimental effects, we do see performance plateauing at iteration 5 on some benchmarks.

---

> ### Author Response · Authors · 2025-06-07
>
> Dear Reviewer, Thank you again for your valuable feedback. As the discussion deadline is June 10, we wanted to ensure that we have addressed all your questions. We welcome any additional thoughts you may have.

---

### Decision · Program_Chairs · 2025-07-08

**Decision:**

Accept

**Comment:**

The paper presents a contribution to the field of video understanding by proposing VideoSAVi, a robust and effective self-training framework for Video-LLMs. The core idea of self-alignment through a self-critiquing mechanism and DPO is well-executed. The empirical results are strong and consistently demonstrate improvements across various benchmarks and backbone models. The authors' comprehensive responses to reviewer questions, including additional experiments and clarifications, should be added to the revised version, particularly regarding comparisons with state-of-the-art models, robustness to noise, and potential overfitting.